# mRNA lipid nanoparticle-mediated pyroptosis sensitizes immunologically cold tumors to checkpoint immunotherapy

Fengqiao Li[1], Xue-Qing Zhang [2,3] ✉, William Ho[1], Maoping Tang[2,3], Zhongyu Li[1], Lei Bu[4] & Xiaoyang Xu [1,5] ✉

Synergistically improving T-cell responsiveness is promising for favorable therapeutic outcomes in immunologically cold tumors, yet current treatments often fail to induce a cascade of cancer-immunity cycle for effective antitumor immunity. Gasdermin-mediated pyroptosis is a newly discovered mechanism in cancer immunotherapy; however, cleavage in the N terminus is required to activate pyroptosis. Here, we report a single-agent mRNA nanomedicine-based strategy that utilizes mRNA lipid nanoparticles (LNPs) encoding only the N-terminus of gasdermin to trigger pyroptosis, eliciting robust antitumor immunity. In multiple female mouse models, we show that pyroptosis-triggering mRNA/LNPs turn cold tumors into hot ones and create a positive feedback loop to promote antitumor immunity. Additionally, mRNA/LNP-induced pyroptosis sensitizes tumors to anti-PD-1 immunotherapy, facilitating tumor growth inhibition. Antitumor activity extends beyond the treated lesions and suppresses the growth of distant tumors. We implement a strategy for inducing potent antitumor immunity, enhancing immunotherapy responses in immunologically cold tumors.

Cancer immunotherapy, especially immune checkpoint blockade (ICB) therapy, is a major therapeutic modality and has prolonged the overall survival in many cancers[1–4]. However, only a minority of patients experience a complete response to immunotherapy (10–30% in solid tumors)[5–7], in part because of the highly immunosuppressive tumor microenvironment (TME) in immunologically cold tumors[8–10]. Synergistically eliciting T-cell immune responses with inflammatory cytokines or immune agonists in cancer immunotherapy is a promising strategy to relieve immunosuppression and activate T cells[11–13]. However, effective antitumor immunity requires activating all the steps of the cancer-immunity cycle, including immunogenic cell death (ICD), maturation of antigen-presenting cells (such as dendritic cells (DCs)), priming and activation of T cells, recruitment of tumor-infiltrating immune cells, and

production of inflammatory cytokines[14–17]. Unfortunately, even combinational therapies using multiple agents often lead to failure in the inhibition of cancer cells. There is thus an unmet need for new therapeutic strategies to induce efficient antitumor immunity and broaden the scope of immunotherapy.

Herein, we propose a single-agent mRNA-based pyroptosis nanomedicine approach that initiates the cancer-immunity cycle and turns cold tumors into inflammatory cytokine-expressing and T cell-infiltrated hot tumors to effectively treat immunologically cold tumors. Pyroptosis is a type of inflammatory programmed cell death that is triggered by the proteolytic cleavage of gasdermin (GSDM) family proteins[18]. The GSDMs are normally self-inhibited through the intramolecular interaction of their N-terminal and C-terminal domains. Upon cleavage by specific caspases and other proteases in the linker

[1]Department of Chemical and Materials Engineering, New Jersey Institute of Technology, Newark, NJ, USA. [2]Shanghai Frontiers Science Center of Drug Target Identification and Delivery, School of Pharmacy, Shanghai Jiao Tong University, Shanghai, PR China. [3]National Key Laboratory of Innovative Immunotherapy, Shanghai Jiao Tong University, Shanghai, PR China. [4]Department of Medicine, NYU Grossman School of Medicine, New York, NY, USA. [5]Department of Biomedical Engineering, New Jersey Institute of Technology, Newark, NJ, USA. ✉e-mail: xueqingzhang@sjtu.edu.cn; xiaoyang.xu@njit.edu

region, the necrotic N-terminal domain form oligomers and translocate to the plasma membrane. The free N-terminal domain binds to lipid components and forms pores in the cell membrane, resulting in rapid plasma membrane rupture and release of danger-associated molecular patterns (DAMPs) and proinflammatory cytokines[19–23]. Immune cells recognize certain DAMPs and then trigger a series of immune responses, including the activation and infiltration of immune cells[23–25]. Additionally, released proinflammatory cytokines through the pyroptotic pore contribute to reversing the immunosuppressive TME. Although these encouraging discoveries, low GSDM expression in many cancers[19, 20] and the complex cleavage process prevent delivering proteases to trigger pyroptosis for antitumor immunity. As such, we hypothesize that pyroptosis induced by direct delivery of the N-terminal GSDM domain is an effective approach to elicit a series of events in the cancer-immunity cycle and transform cold tumors into hot tumors.

mRNA nanomedicine-based gene therapy represents a promising therapeutic strategy for multiple clinical applications. Recently, our group successfully developed a formulation to synthesize ionizable cationic lipid nanoparticles (LNPs), termed AA3-Dlin LNPs, with good safety and high mRNA translation efficiency in vitro and in vivo[26]. It holds great promise to synergistically combine mRNA/LNPs with GSDM for cancer treatment by triggering pyroptosis. Herein, we present an mRNA-based nanomedicine approach where the AA3-Dlin LNP formulation encapsulates a single-agent mRNA encoding the GSDMB N-terminal domain, termed GSDMB^NT mRNA@LNPs. The developed GSDMB^NT mRNA@LNP formulation is self-assembled by an ionizable cationic lipid (AA3-Dlin), phospholipid (DOPE), cholesterol, and PEG, and GSDMB^NT mRNA is encapsulated inside LNPs via electrostatic interactions. We expect that this LNP formulation can be delivered into tumor tissue where the mRNA is translated into the N-terminal domain of GSDMB protein, triggering pyroptosis directly without protease cleavage. Pyroptosis has the capacity to induce immunologic cell death (ICD), initiate the release of proinflammatory cytokines, as well as to activate and recruit immune cells within tumors, which in turn leads to a cascade of events that further promotes cell death, cytokine release, and activation of immune responses. The resulting positive feedback loop can create a favorable immunogenic hot tumor microenvironment that sensitizes cancer cells to ICB-mediated immunotherapy, showing superior tumor inhibition compared with monotherapy (Fig. 1a).

Indeed, our in vitro results demonstrate that even low levels of tumor cell pyroptosis triggered by single-agent GSDMB^NT mRNA@LNPs are sufficient to induce robust ICD. In multiple immunologically cold tumor models, our results show that pyroptosis-triggering mRNA/LNPs can inhibit tumor growth and extend overall survival, accompanied by stimulation of proinflammatory cytokines and promotion of the recruitment of immune cells in the TME. Moreover, pyroptosis-triggering mRNA/LNPs can improve the therapeutic benefits of immune checkpoint inhibitor (anti-programmed death-1 antibody, aPD-1)-mediated immunotherapy and even achieve tumor elimination and long-term survival in both orthotopic 4T1 breast carcinoma and highly aggressive B16F10 melanoma models. In addition, we find that pyroptosis-triggering mRNA/LNPs can potently synergize with aPD-1-mediated immunotherapy, induce a local immune response and subsequently provoke a systemic effect, to eradicate large melanomas and inhibit the growth of distant tumors in a B16F10 dual-tumor model. Collectively, our single-agent pyroptosis-triggering mRNA/LNPs approach offers a facile and highly efficacious strategy to achieve potent antitumor immunity and enhance immunotherapy in immunologically cold tumors, and the method described here provides a versatile platform that can be potentially extended to other immunotherapies besides aPD-1, holding a high translational promise.

## Results

### Preparation and characterization of GSDMB^NT mRNA@LNPs

GSDMB^NT mRNA was first synthesized using an in vitro transcription method[27] with significant cap modification and sequence optimization. An agarose gel assay was performed to confirm the size of synthesized GSDMB^NT mRNA (Supplementary Fig. 1). Effective mRNA delivery relies on the physical and chemical characteristics of formulations. Therefore, an AA3-Dlin LNP platform was then prepared to encapsulate GSDMB^NT mRNA (GSDMB^NT mRNA@LNPs), as previously reported[26]. The morphology, hydrodynamic diameter, and zeta potential of GSDMB^NT mRNA@LNPs were evaluated by transmission electron microscopy (TEM) and dynamic light scattering (DLS), respectively. As shown in Fig. 1b, the TEM analysis revealed that GSDMB^NT mRNA@LNPs are spherical in morphology and have a smooth surface. The particle size, polydispersity index (PDI), and zeta potential of GSDMB^NT mRNA@LNPs as assessed from DLS was about $119.2 \pm 1.943$ nm, $0.104 \pm 0.022$ and $-0.102 \pm 0.595$ mV, respectively (Fig. 1c). The stability of the developed GSDMB^NT mRNA@LNPs was evaluated by monitoring particle sizes in various culture media, including pH 6.5 buffer (to mimic the weakly acidic pH of many tumors), pH 7.4 buffer (to mimic the normal physiological pH), 10% or 20% plasma (to mimic esterase-enriched conditions). GSDMB^NT mRNA@LNPs exhibit no obvious changes in particle sizes when incubated under the conditions indicated, implying good stability within acidic or esterase-enriched pathological tumor microenvironments (Fig. 1d and Supplementary Fig. 2).

Next, we examined the intracellular uptake of the FITC-labeled LNPs (Luc mRNA@^FITC LNPs) in four types of cells by confocal laser scanning microscopy (CLSM). Figure 1e shows that LNPs could be efficiently taken up by various cell types and the intracellular 488 nm green signal derived from ^FITC LNPs increased proportionally with the incubation time, demonstrating a time-dependent cellular internalization. We also formulated Luc ^Cy5 mRNA-loaded LNPs (termed Luc ^Cy5 mRNA@LNPs) and studied their endosomal escape ability. Figure 1f and Supplementary Fig. 3 revealed that the majority of the red signal from ^Cy5 mRNA was not colocalized with the green signal from the endosomes or lysosomes following 4 h of incubation, suggesting the successful cytosolic release of mRNA mediated by LNPs.

The transfection efficiency of AA3-Dlin LNPs was tested by encapsulating mRNA encoding mCherry red fluorescent protein (termed mCherry mRNA@LNPs) and tested in HEK 293, HeLa, 4T1, and B16F10 cells. As shown in Supplementary Fig. 4, the mCherry-loaded LNPs showed transfection efficacies of 83.5%, 75.6%, 46.8%, and 45.3% in HEK 293, HeLa, 4T1 and B16F10 cells, respectively. Western blot was performed to confirm the expression of GSDMB N-terminal domain in cells transfected with GSDMB^NT mRNA@LNPs (Supplementary Fig. 5).

### GSDMB^NT mRNA@LNPs induce immunogenic pyroptosis

When treated with GSDMB^NT mRNA@LNPs, pyroptotic morphological changes involving cytoplasmic swelling and membrane rupture were observed in HEK 293, HeLa, 4T1, and B16F10 cells (Fig. 2a and Supplementary Fig. 6-7). These phenomena were not observed in the control groups including naked GSDMB^NT mRNA and LNP-treated groups. The lactate dehydrogenase (LDH) release assay was used to evaluate the lethal effect of GSDMB^NT mRNA@LNPs on cancer cells. As shown in Fig. 2b, GSDMB^NT mRNA@LNP treatment led to markedly increased cell death rates of 51.5%, 47.5%, 32.6%, and 23.3% for HEK 293, HeLa, 4T1, and B16F10 cells, respectively, compared to the control groups (naked GSDMB^NT mRNA or blank LNPs) which exhibited substantially lower rates of below 5% for all four cell types. The cell viability also was measured by calcein release study (Supplementary Fig. 8). Then, Annexin V/propidium iodide (PI) apoptosis assay was used to quantitate the lethal effect of GSDMB^NT mRNA@LNPs on different cells. Supplementary Fig. 9 reveals concentration-dependent and time-dependent apoptosis of HEK 293 cells treated with GSDMB^NT

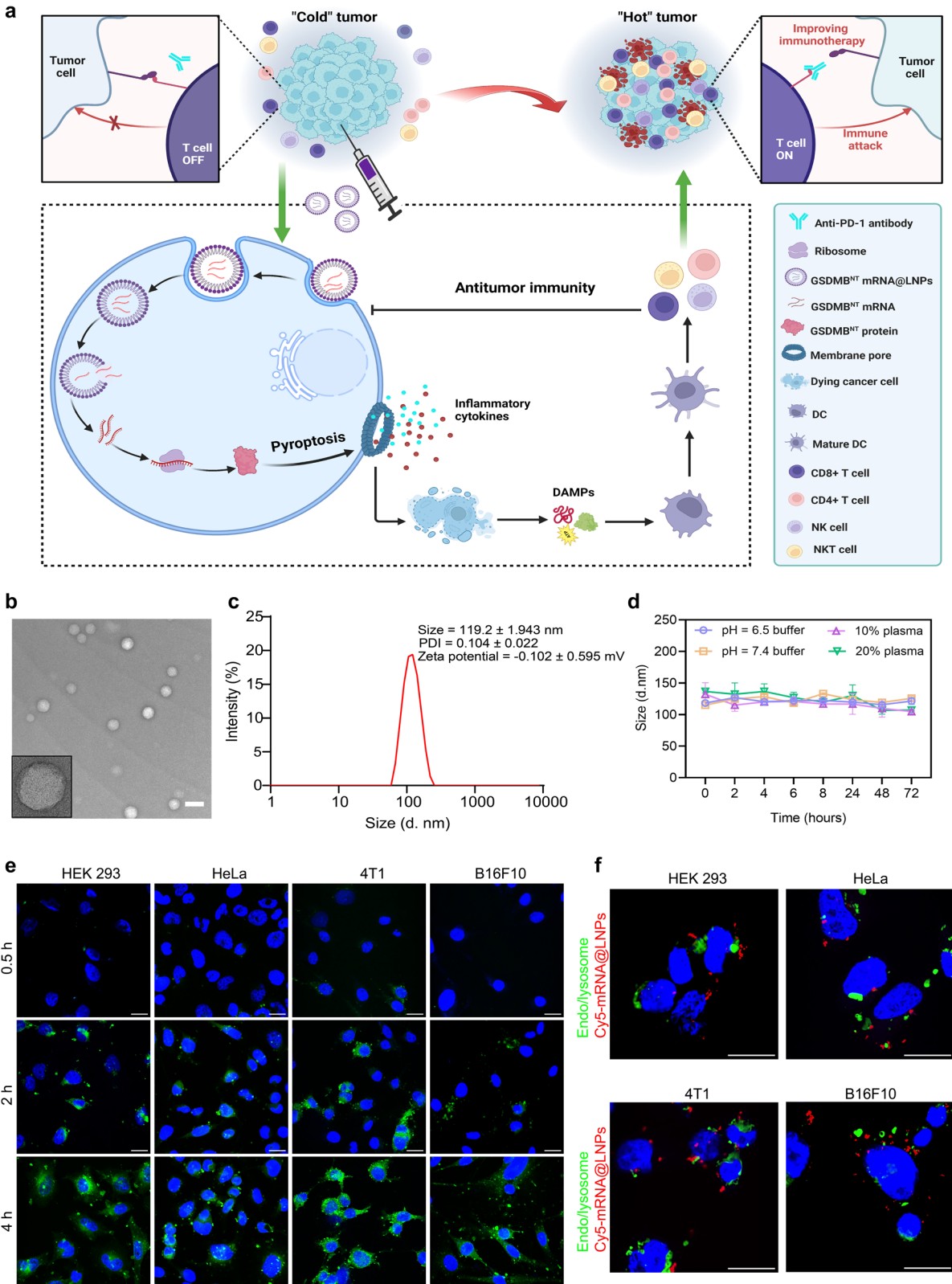

mRNA@LNPs. About 40% of HEK 293 cells underwent apoptosis after 24 h of treatment with 1.0 µg/mL GSDMB$^{NT}$ mRNA@LNPs, which indicates that a timeframe of 24 h and dosage of 1.0 µg/mL of treatment is sufficient to trigger apoptosis. Under the same treatment time and dose conditions, about 30%, 25%, and 20% of HeLa, 4T1, and B16F10 cells underwent apoptosis after GSDMB$^{NT}$ mRNA@LNP treatment (Fig. 2c).

## GSDMB$^{NT}$ mRNA@LNPs promote immunogenic cell death (ICD) and stimulate DC maturation

Pyroptosis promotes the release of DAMPs to stimulate immune responses[28], thus we hypothesized that GSDMB$^{NT}$ mRNA@LNPs might induce the release of DAMPs to promote immunogenic cell death (ICD). To confirm this, the expression of ICD biomarkers were measured in cells, including surface-exposed calreticulin (CRT) ("eat" me

**Fig. 1 | Schematic of antitumor immunity via GSDMB^NT mRNA@LNP-mediated pyroptosis and characterization of GSDMB^NT mRNA@LNPs. a** Intratumoral administration of mRNA lipid nanoparticles encoding only the N-terminal domain of GSDMB triggers pyroptosis, eliciting antitumor immunity and facilitating anti-PD-1-mediated immunotherapy in immunologically cold tumors. **b** TEM image of GSDMB^NT mRNA@LNPs. Scale bar = 200 nm. **c** The particle size, PDI, and zeta potential of GSDMB^NT mRNA@LNPs were analyzed by DLS. Data are representative of four independent experiments. **d** DLS measurement of GSDMB^NT mRNA@LNPs under the conditions indicated, including pH 6.5 buffer, pH 7.4 buffer, 10% or 20%

plasma. Data are presented as means ± SD (n = 3). **e** Cellular uptake of FITC-labeled LNPs (Luc mRNA@^FITC LNPs) was monitored in HEK 293, HeLa, 4T1, and B16F10 cells at different time points. Scale bar = 20 μm. **f** LNP-mediated endosomal/lysosomal escape and cytoplasmic release of Luc ^Cy5 mRNA in HEK 293, HeLa, 4T1, and B16F10 cells 4 h after incubation. DAPI (blue), Endo/lysosome (green), Luc ^Cy5 mRNA@LNPs (red), scale bar = 20 μm. Data shown in **e**, **f** are representative of two independent experiments. Source data are provided as a Source Data file. Cartoon in panel **a** was created with BioRender.com.

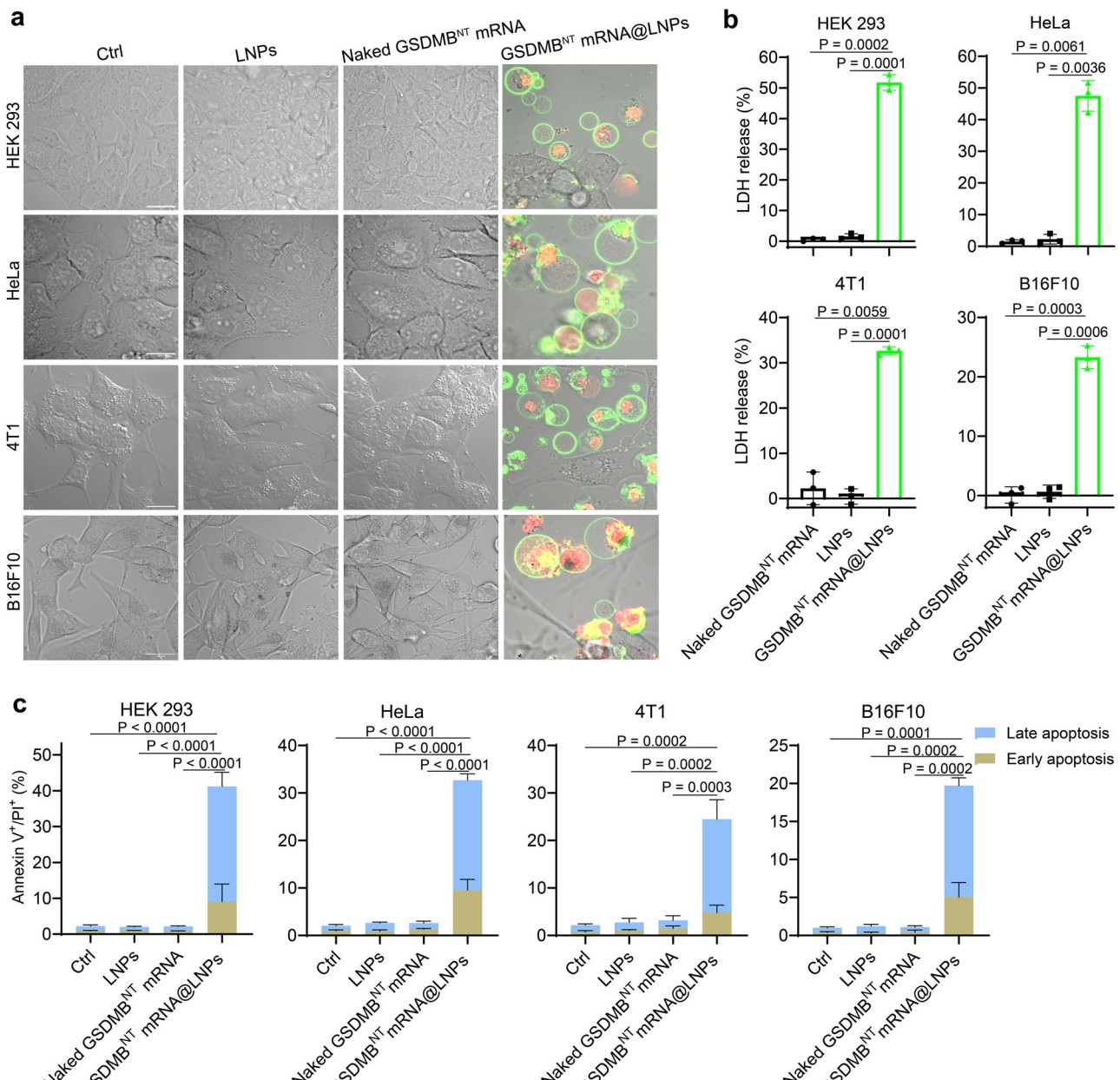

**Fig. 2 | Lipid nanoparticles deliver mRNA encoding the N-terminal domain of GSDMB into cells to induce pyroptosis. a** Cell morphologies of the treated HEK 293, HeLa, 4T1 and B16F10 cells were detected using a confocal microscope. Before imaging, cells were treated with annexin V-FITC and propidium iodide (PI) and incubated for 15 min. Scale bars = 20 μm. Data are representative of three independent experiments. **b** LDH release-based cell death assay in HEK 293, HeLa, 4T1, and B16F10 cells after treatment with naked GSDMB^NT mRNA, LNPs, or GSDMB^NT

mRNA@LNPs, respectively. Data are presented as means ± SD (n = 3). Statistical significance was calculated via one-way ANOVA. **c** Flow-cytometry analysis of cells positive for propidium iodide and annexin V. Data are presented as means ± SD (n = 3). Statistical significance was calculated using a two-tailed Student's t test. Untreated cells served as the control (Ctrl) in all experiments. Source data are provided as a Source Data file.

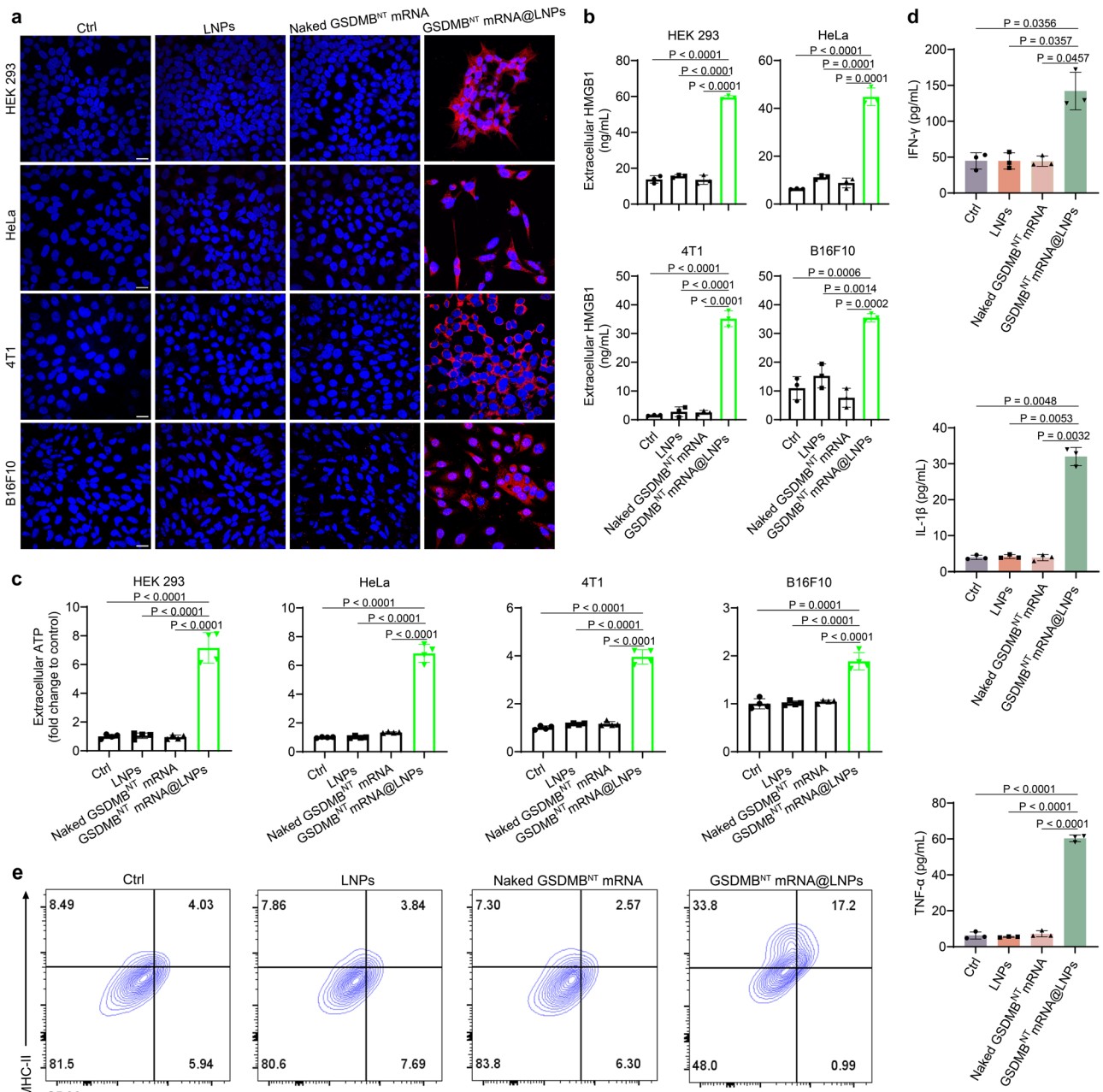

**Fig. 3 | GSDMB^NT mRNA@LNPs induce immunogenic pyroptosis and stimulate the maturation of bone marrow-derived dendritic cells (BMDCs). a** A confocal microscope was used to detect CRT expression in treated HEK 293, HeLa, 4T1, and B16F10 cells. Scale bar = 20 μm. Data are representative of three independent experiments. **b, c** Extracellular HMGB1 and ATP expression were analyzed by ELISA in HEK 293, HeLa, 4T1, and B16F10 cells after different treatments. Data in **b** are presented as means ± SD (n = 3). Data in **c** are presented as means ± SD (n = 4). Statistical significance was calculated using a two-tailed Student's t test.

**d, e** Immune stimulation of BMDCs by GSDMB^NT mRNA@LNPs. B16F10 cells were pretreated with PBS (Ctrl), naked GSDMB^NT mRNA, blank LNPs or GSDMB^NT mRNA@LNPs, followed by coculture with BMDCs for 48 h. **d** Quantitative determination of proinflammatory cytokines IFN-γ, IL-1β and TNF-α using ELISA assay. Data are presented as means ± SD (n = 3). Statistical significance was calculated via one-way ANOVA. **e** Analysis of DC maturation biomarkers (MHC-II and CD86) using flow cytometry (n = 3). Source data are provided as a Source Data file.

signal), extracellularly released high mobility group box 1 (HMGB1) ("danger" signal), and adenosine triphosphate (ATP) ("find me" signal)[29]. As shown in Fig. 3a, cells incubated with LNPs or naked GSDMB^NT mRNA had almost no CRT expression, while significant CRT signals were observed in GSDMB^NT mRNA@LNPs transfected cells. The flow cytometry analysis also showed a significant increase in the percentage of CRT+ cells after GSDMB^NT mRNA@LNP treatment compared to control groups (naked GSDMB^NT mRNA or blank LNPs) (Supplementary Fig. 10). Compared to controls, HMGB1 and ATP levels

were significantly increased in all four cell lines after treatment with GSDMB^NT mRNA@LNPs for 48 h as determined by ELISA (Fig. 3b, c).

To explore the immune stimulation of DCs induced by GSDMB^NT mRNA@LNP-mediated pyroptosis, B16F10 cells were pretreated with PBS (Ctrl), naked GSDMB^NT mRNA, blank LNPs or GSDMB^NT mRNA@LNPs, followed by coculture with BMDCs obtained from female C57BL/6 mice. As shown in Fig. 3d, e and Supplementary Fig. 11, pretreatment with GSDMB^NT mRNA@LNPs increased production of IFN-γ, IL-1β and TNF-α by 3.2-, 8.1-, and 9.6-fold, respectively, over PBS

pretreatment (Ctrl). Additionally, coculture with GSDMB^NT mRNA@LNP-pretreated B16F10 cells resulted in a significant upregulation of CD86 and MHC-II surface expression on BMDCs 1.8- and 4.1-fold, respectively, compared to the control group. Taken together, pyroptotic tumor cells efficiently induce the maturation of DCs via GSDMB^NT mRNA/LNP-mediated pyroptosis.

Cell Counting Kit-8 (CCK-8) assay demonstrated limited cytotoxicity in bone marrow-derived macrophages and DCs after treatment with GSDMB^NT mRNA@LNPs at an mRNA concentration higher than 1.5 μg/mL (Supplementary Fig. 12).

### Treatment with GSDMB^NT mRNA@LNPs improves tumor control in an aPD-1-resistant 4T1 breast cancer mouse model

As reported previously, mice bearing 4T1 breast carcinoma are resistant to immune checkpoint inhibitors, such as anti-PD-1 (aPD-1) therapy[30]. To investigate the translation efficacy of synthesized mRNA after intratumoral administration, we transcribed two kinds of firefly luciferase-encoding mRNAs (Luc mRNAs) with different capping methods, Anti-Reverse Cap Analog (ARCA)-capped Luc mRNA and CleanCap-capped Luc mRNA. Subsequently, to evaluate the in vivo transfection effect of LNPs, these two engineered mRNAs were encapsulated into LNPs to form ARCA-capped Luc mRNA@LNPs or CleanCap-capped Luc mRNA@LNPs. The fabricated Luc mRNA@LNPs were intratumorally injected into an orthotopic 4T1 breast tumor model and luciferase activity was analyzed by in vivo bioluminescent imaging after 6 hours. As shown in Supplementary Fig. 13a, a four-times higher bioluminescence intensity was observed in the tumors treated with CleanCap-capped Luc mRNA@LNPs. Almost no luciferase signal was detected in major organs, which indicated that the majority of the Luc mRNA@LNPs had accumulated in the tumors. Thus, GSDMB^NT mRNA@LNPs with CleanCap-capping were used for further animal experiments. Moreover, a single injection of CleanCap-capped Luc mRNA@LNPs maintained elevated levels of the bioluminescence signal for 3 days in 4T1 tumor sites (Supplementary Fig. 13b). These findings suggest that our mRNA/LNPs delivery system is capable of transfecting cells for a fast, robust, and durable gene expression in vivo.

To explore the therapeutic efficacy of intratumorally administered GSDMB^NT mRNA@LNPs, we first evaluated antitumor immune responses in the orthotopic 4T1 breast tumor model. A total 40 μg of GSDMB^NT mRNA@LNPs was intratumorally injected into 4T1 tumors and then the concentrations of inflammatory cytokines including tumor necrosis factor-alpha (TNF-α) and interferon-gamma (IFN-γ) in tumor tissue or serum were assayed by ELISA after 6, 24, or 72 h, separately (Fig. 4a). Both TNF-α and IFN-γ in serum and tumor increased after 6 h of GSDMB^NT mRNA@LNP treatment and maintained elevated levels for 3 days. This result inspired us to further explore the antitumor efficacy of GSDMB^NT mRNA@LNPs in vivo.

To further investigate whether GSDMB^NT mRNA@LNPs could sensitize immunological cold tumors to ICB-mediated therapy, we evaluated the antitumor activity of combinatorial treatments of GSDMB^NT mRNA@LNPs and aPD-1 in orthotopic 4T1 breast cancer models following the treatment timeline in Fig. 4b. Briefly, wild-type female Balb/c mice were inoculated subcutaneously with ~$5 \times 10^5$ 4T1 tumor cells into the fourth mammary fat pad and then intratumorally administered on days 7, 10, 13, and 16 with PBS, LNPs, or GSDMB^NT mRNA@LNPs (with 10 μg GSDMB^NT mRNA). aPD-1-treated mice received intraperitoneal administration of anti-PD-1 antibodies (100 μg per mouse) on days 8, 11, 14, and 17. Tumor growth was recorded by tumor volume measurements taken every two days and the survival monitoring ended at day 48 (30 days after the final treatment). After four rounds of treatments, our results showed that tumors grew rapidly in PBS, LNPs, and aPD-1-treated groups, and all mice died within 37 days, indicating that aPD-1 did not inhibit tumor growth in 4T1 tumors. In contrast, GSDMB^NT mRNA@LNPs effectively inhibited tumor growth ($P = 0.0018$) and prolonged animal survival to 45 days.

Massive tumor shrinkage ($P = 0.0005$) and a more than 70% survival rate ($P = 0.0004$) occurred in mice treated with the GSDMB^NT mRNA@LNPs + aPD-1 combination therapy at day 48 (Fig. 4c–e). These results indicate that GSDMB^NT mRNA@LNPs improved tumor suppression and strengthened the sensitivity of aPD-1 antibody therapy in an aPD-1-resistant 4T1 tumor model.

### GSDMB^NT mRNA@LNPs induce ICD via pyroptosis in an anti-PD-1-resistant 4T1 breast cancer mouse model

To confirm that the antitumor immunity in the 4T1 breast cancer mouse model was induced by GSDMB^NT mRNA@LNP-mediated pyroptosis, we investigated the expression of ICD biomarkers (CRT and HMGB1) and pyroptosis-related cytokines (IL-1β and IL-18). The immunofluorescence analysis of tumor tissues showed increased CD8 and CRT expression in the GSDMB^NT mRNA@LNP group compared with the control groups. After the combination treatment of GSDMB^NT mRNA@LNPs and aPD-1, CD8 and CRT expression increased by 6.7- and 4.3-fold compared with the PBS-treated control group. Almost no changes in these biomarkers were detected in the group of aPD-1 treatment alone (Fig. 5a, b). Supplementary Fig. 14 illustrates that in comparison to control groups, both GSDMB^NT mRNA@LNPs and combination treatment groups significantly increased surface exposure of CRT in 4T1 tumor tissues, with higher CRT surface exposure achieved in the group receiving the combination therapy treatment. The evaluation of PI-positive cells can indicate dead cells, and in tumor tissues we indeed found increased dead cells in the GSDMB^NT mRNA@LNP treatment and GSDMB^NT mRNA@LNPs + aPD-1 combination treatment groups (Supplementary Fig. 15a). The ELISA results illustrate the robust induction of IL-1β, IL-18 and HMGB1 production, an indicator of pyroptosis-induced ICD, in both tumor and serum samples following treatment with GSDMB^NT mRNA@LNPs alone or in combination with a-PD1 (Fig. 5c).

To assess the in vivo safety profile of GSDMB^NT mRNA@LNPs, we monitored body weight changes and performed aminotransaminase analysis following the administration of LNPs. As shown in Supplementary Fig. 16, the healthy mice treated with GSDMB^NT mRNA@LNPs show no change in body weight. Quantitative determination of major liver function markers (alanine transaminase (ALT) and aspartate transaminase (AST)) demonstrated that administered GSDMB^NT mRNA@LNPs did not induce any obvious hepatic dysfunction (Fig. 5d). These results collectively illustrate that GSDMB^NT mRNA@LNPs trigger antitumor immunity in a safe manner.

### Treatment with GSDMB^NT mRNA@LNPs remodels the tumor microenvironment and enhances potent antitumor activity in an aggressive B16F10 melanoma mouse model

To further verify the synergistic effect of GSDMB^NT mRNA@LNPs in immunotherapy, we investigated the antitumor activity in an aggressive melanoma mouse model. GSDMB^NT mRNA@LNPs and aPD-1 antibodies were administered in B16F10 tumor models following the timeline as shown in Fig. 6a. Briefly, $5 \times 10^5$ B16F10-Luc cells were implanted subcutaneously on the right flank of the C57BL/6 female mice to establish subcutaneous tumors. After 7 days of cell implantation, mice were intratumorally injected with PBS, LNPs, or GSDMB^NT mRNA@LNPs (with 10 μg GSDMB^NT mRNA) on days 7, 10, 13, and 16. aPD-1-treated mice received intraperitoneal administration of anti-PD-1 antibodies (100 μg per mouse) on days 8, 11, 14, and 17. Tumor imaging was carried out every 5 days for a total of 4 times from initial treatment and survival analysis ended at day 58 (40 days after the final treatment). In Fig. 6b, c, the images and quantitative analysis of bioluminescence signals showed that tumors grew rapidly in control groups, and 3/8 (PBS-treated) and 2/7 (LNP-treated) mice died within 23 days. The aPD-1 alone treatment exhibited an antitumor effect at early time points but failed to achieve sustained tumor inhibition. The survival time of aPD-1-treated mice was slightly prolonged from 32 days in PBS and LNP-treated mice to 36 days, indicating that aPD-1 is not effective

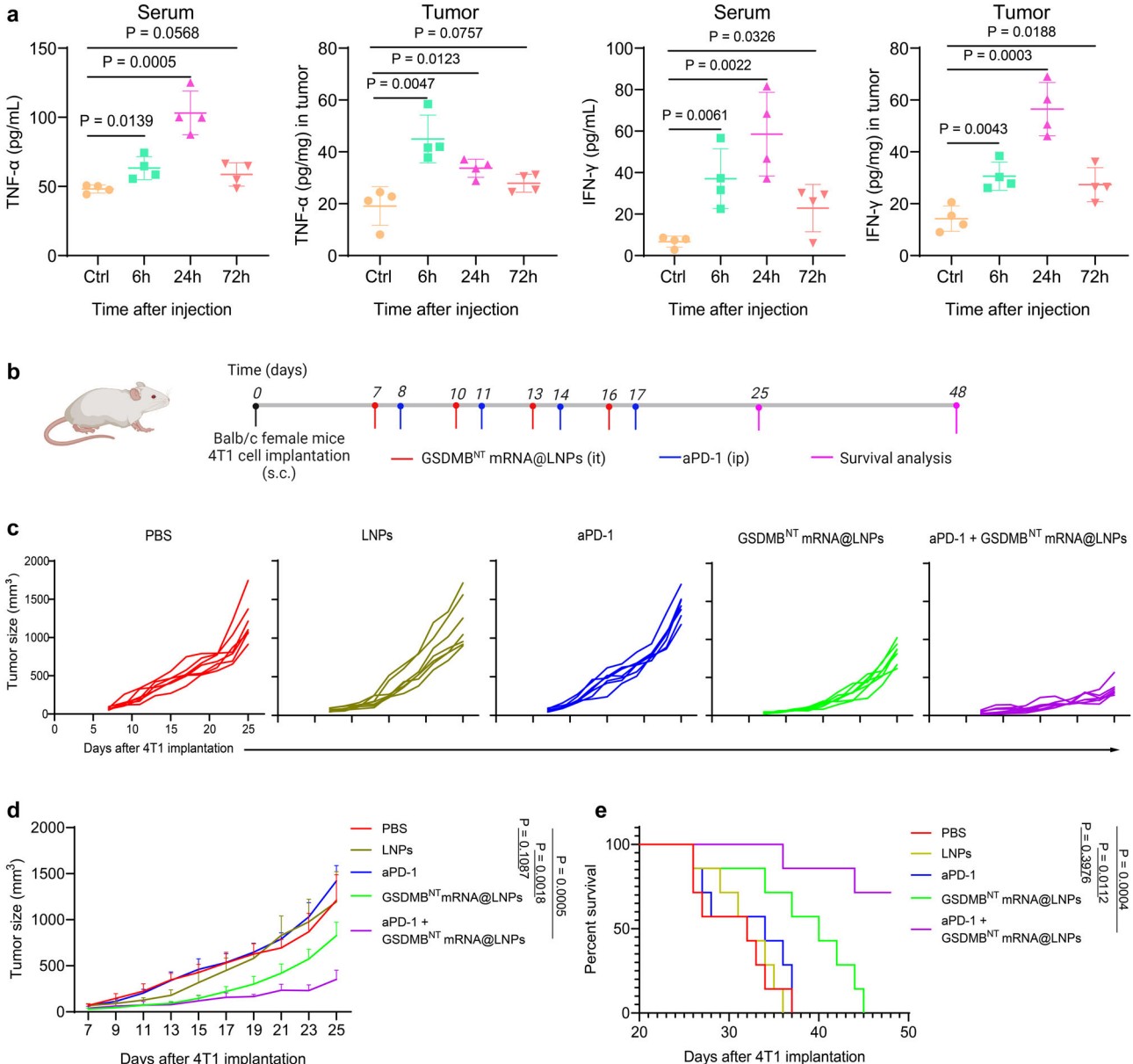

**Fig. 4 | Treatment with GSDMB^NT mRNA@LNPs promotes tumor control in an anti-PD-1-resistant 4T1 breast cancer mouse model. a** 6, 24, and 72 h after a single dose of GSDMB^NT mRNA@LNPs (40 μg mRNA), cytokine concentrations were measured in tumor tissue or serum by ELISA. Data are presented as means ± SD (*n* = 4 mice per group). Statistical significance was calculated using a two-tailed Student's *t* test. **b** Experimental timeline for treatment of 4T1 orthotopic tumor-bearing mice. s.c., subcutaneous; it, intratumoral; ip, intraperitoneal. **c** Individual

growth curves of tumor size for mice treated as indicated. **d, e** The average tumor growth curves and survival percentages for mice treated as indicated (*n* = 7 mice per group). Data shown in **e** are represented as means ± SD. *P* values were determined by two-tailed unpaired Student's *t* test in **c, d** or by log-rank (Mantel-Cox) test in **e**. Source data are provided as a Source Data file. Panel **b** was created with BioRender.com.

in suppressing tumors. In contrast, mice treated with GSDMB^NT mRNA@LNPs showed superior tumor inhibition and extended survival time (52 days). Especially, the combinational therapy of GSDMB^NT mRNA@LNPs and aPD-1 showed that established tumors were eliminated in 7 of 10 mice and 70% of mice still survived (*P* < 0.0001) at the predetermined endpoint (day 58) (Fig. 6d). Consistently, compared to mice treated with aPD-1 alone, increased CD8, CRT, and HMGB1 expression and PI-positive cells were measured in mice treated with combinational therapy (Fig. 6e–h and Supplementary Fig. 15b). The flow cytometry analysis also revealed a notable upregulation of CRT+ cells in tumor tissues from B16F10 tumor-bearing mice receiving a combined treatment of a-PD1 and GSDMB^NT mRNA@LNPs (Supplementary Fig. 17). As shown in Supplementary Fig. 18, the proposed

combination therapy results in a higher level of cell death in tumor tissues compared to the PBS control. These findings demonstrate that GSDMB^NT mRNA@LNPs play an important role in improving the therapeutic efficacy of aPD-1-mediated immunotherapy.

To better understand how the GSDMB^NT mRNA@LNPs remodel the tumor microenvironment, as shown in Fig. 7a, tumors and blood were harvested to assess the concentrations of cytokines on day 18. As shown in Fig. 7b, mice treated with PBS and LNPs alone had similar concentrations of TNF-α and IFN-γ in serum and tumor, while the concentrations of cytokines significantly increased in mice with GSDMB^NT mRNA@LNP treatments. Furthermore, we identified and characterized different immune cell populations in lymph nodes and tumors using flow cytometry. As shown in Fig. 7c and Supplementary

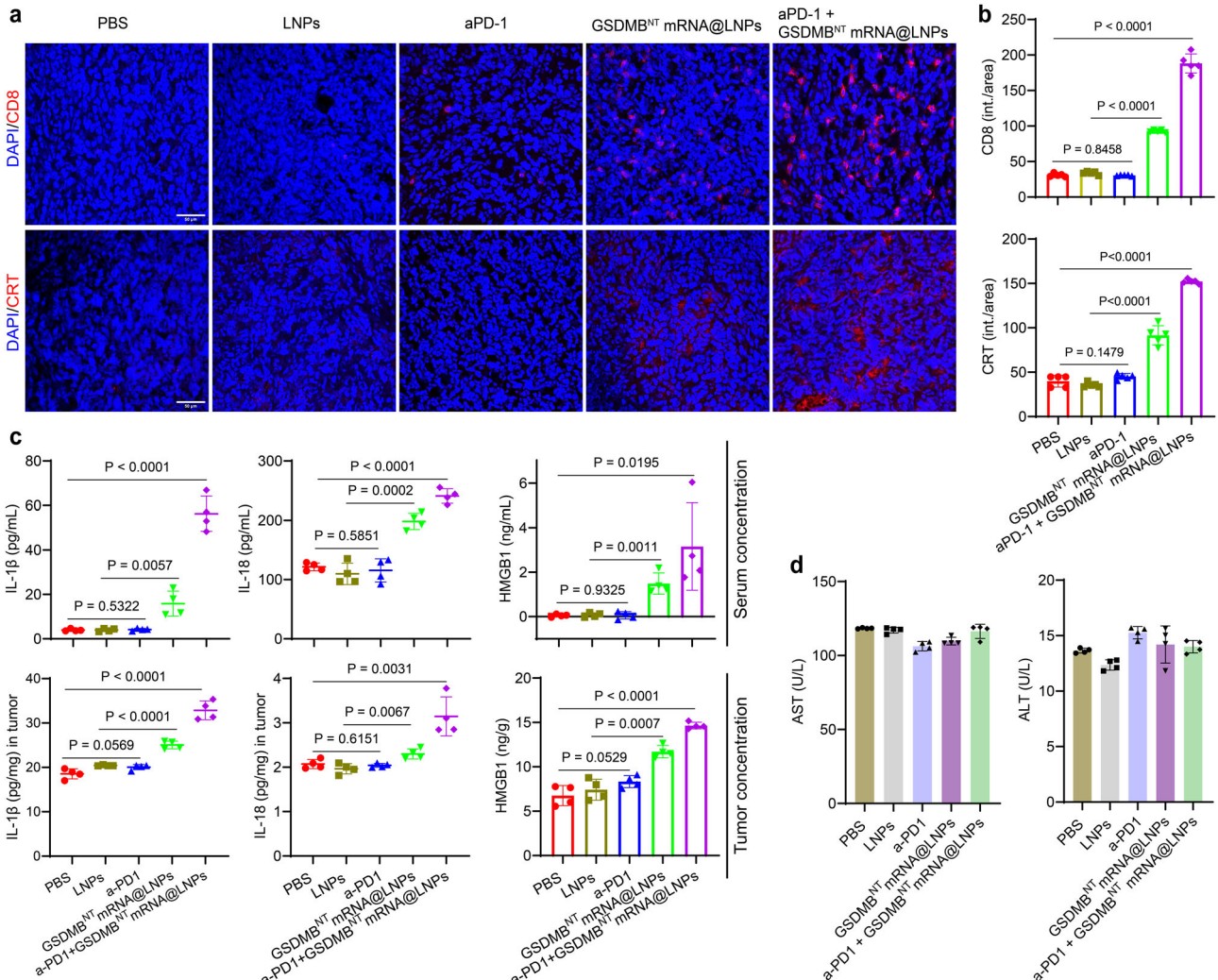

**Fig. 5 | Treatment with GSDMB^NT mRNA@LNPs induces antitumor immunity in an anti-PD-1-resistant 4T1 breast cancer mouse model. a** Immunofluorescence staining of tumors for CD8⁺ T cell infiltration and CRT expression after indicated treatments. Scale bar = 50 μm. **b** Quantitative analysis of immunofluorescence staining in terms of CD8⁺ and CRT intensities. Data are presented as means ± SD (*n* = 5). Statistical significance was calculated using a two-tailed Student's *t* test. **c** ELISA analysis of IL-1β, IL-18, and HMGB1 in tumor and serum samples from 4T1 tumor-bearing mice receiving the treatments indicated. Data are presented as means ± SD (*n* = 4 mice per group). Statistical significance was calculated using a two-tailed Student's *t* test. **d** Blood samples were collected on the second day after the final injection for aminotransaminase analyses. Data are presented as means ± SD (*n* = 4 mice per group). Source data are provided as a Source Data file.

Fig. 19-20, DCs, CD4⁺ T cells, CD8⁺ T cells, natural killer (NK) cells, and NK T cells were recruited in tumors after the administration of GSDMB^NT mRNA@LNPs. Strikingly, compared to the aPD-1 monotherapy group, the population of CD4⁺ T cells showed a 15.6-fold increase in the GSDMB^NT mRNA@LNPs + aPD-1 group. Consistently, the tumor weight of mice treated with the combinational therapy was reduced by 17.9-fold compared to that of mice with aPD-1 treatment (Fig. 7d). Meanwhile, these therapies did not induce body weight loss or cell death and abnormality in major organs (Supplementary Fig. 16 and 21). These results reveal that GSDMB^NT mRNA@LNPs exhibit antitumor immunity by reversing the immunosuppressive TME.

In the in vivo rechallenge study, B16F10 tumor-bearing mice pre-treated with a combined treatment regimen of GSDMB^NT mRNA@LNPs and a-PD1 were resistant to tumor rechalle, which is indicative of the establishment of immunological memory (Supplementary Fig. 22).

### Local administration of GSDMB^NT mRNA@LNPs enhances anti-tumor immunity in distant tumors

To further evaluate whether GSDMB^NT mRNA@LNPs can induce sys-temic immunity against untreated tumors, we established a B16F10

dual-tumor model that was inoculated with B16F10-Luc cells on the left and right flanks. Following the timeline in Fig. 8a, combinational therapy mice were treated with GSDMB^NT mRNA@LNPs (with 10 μg GSDMB^NT mRNA) on days 7, 10, 13, and 16, and treated with aPD-1 (100 μg per mouse) on days 8, 11, 14, and 17. The in vivo bioluminescence imaging in Fig. 8b, c revealed that the combinational treatment of aPD-1 antibody and GSDMB^NT mRNA@LNPs induced regression of treated tumors and inhibited the growth of untreated tumors. CD8 and CRT expression analysis also confirmed the improvement in the immunosuppressive TME by combinational therapy (Fig. 8d, e). We conclude that local GSDMB^NT mRNA@LNP treatment not only triggers inflammatory pyroptosis in tumors but also promotes systemic anti-tumor immunity, which can control tumor growth at remote sites.

## Discussion

Cancer immunotherapy is a major therapeutic modality for the treatment of many cancers. This method aims to activate or boost the ability of endogenous T cells within the tumor to recognize and destroy cancer cells through natural immune mechanisms[1–3, 31]. Particularly, the advent of ICB therapy has greatly advanced the field of

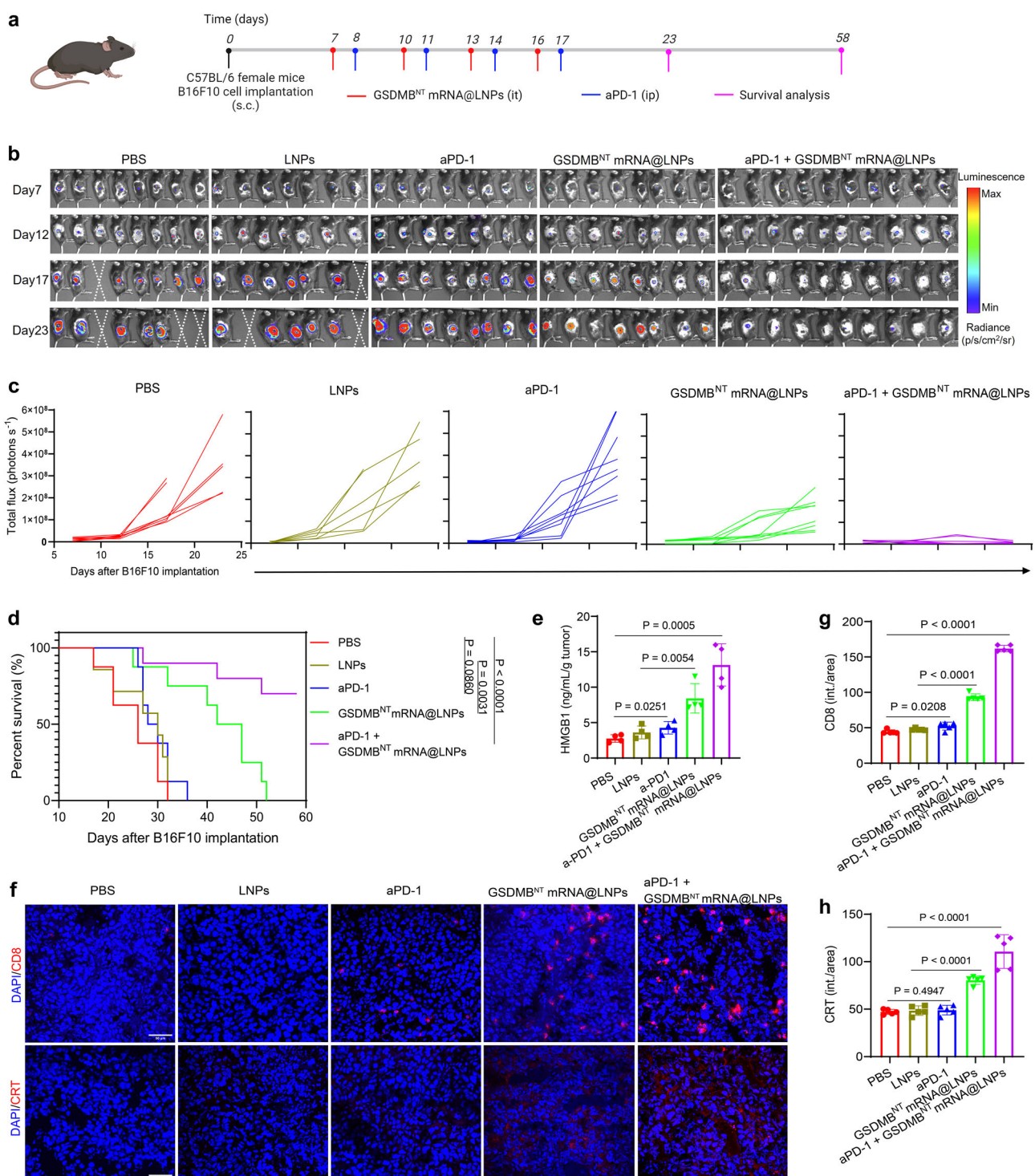

**Fig. 6 | Treatment with GSDMB^NT mRNA@LNPs enhances potent antitumor activity in an aggressive melanoma mouse model. a** Experimental timeline for treatment of B16F10 tumor-bearing mice. **b**, **c** In vivo bioluminescence images and quantification of luciferase signals in mice treated as indicated for monitoring tumor growth. Imaging was performed every 5 days from the initial treatment day (day 7 after tumor inoculation) until day 23. **d** The survival percentages for mice treated as indicated. $n = 8$ mice for PBS, aPD-1, or GSDMB^NT mRNA@LNP treatment groups, $n = 7$ mice for LNP treatment group, and $n = 10$ mice for aPD-1 + GSDMB^NT mRNA@LNP treatment group. Survival analysis was analyzed using the log-rank

(Mantel-Cox) test. **e** ELISA analysis of HMGB1 in the supernatant of B16F10 tumors excised from mice treated as indicated. Data are presented as means ± SD ($n = 4$ mice per group). **f**–**h** Representative images and quantitative analysis of immunofluorescence staining for CD8^+ T cell infiltration and CRT expression in tumors after indicated treatments. Scale bar = 50 μm. Results are presented as means ± SD ($n = 5$). Statistical significance was calculated using a two-tailed Student's $t$ test. Source data are provided as a Source Data file. Panel **a** is created with BioRender.com.

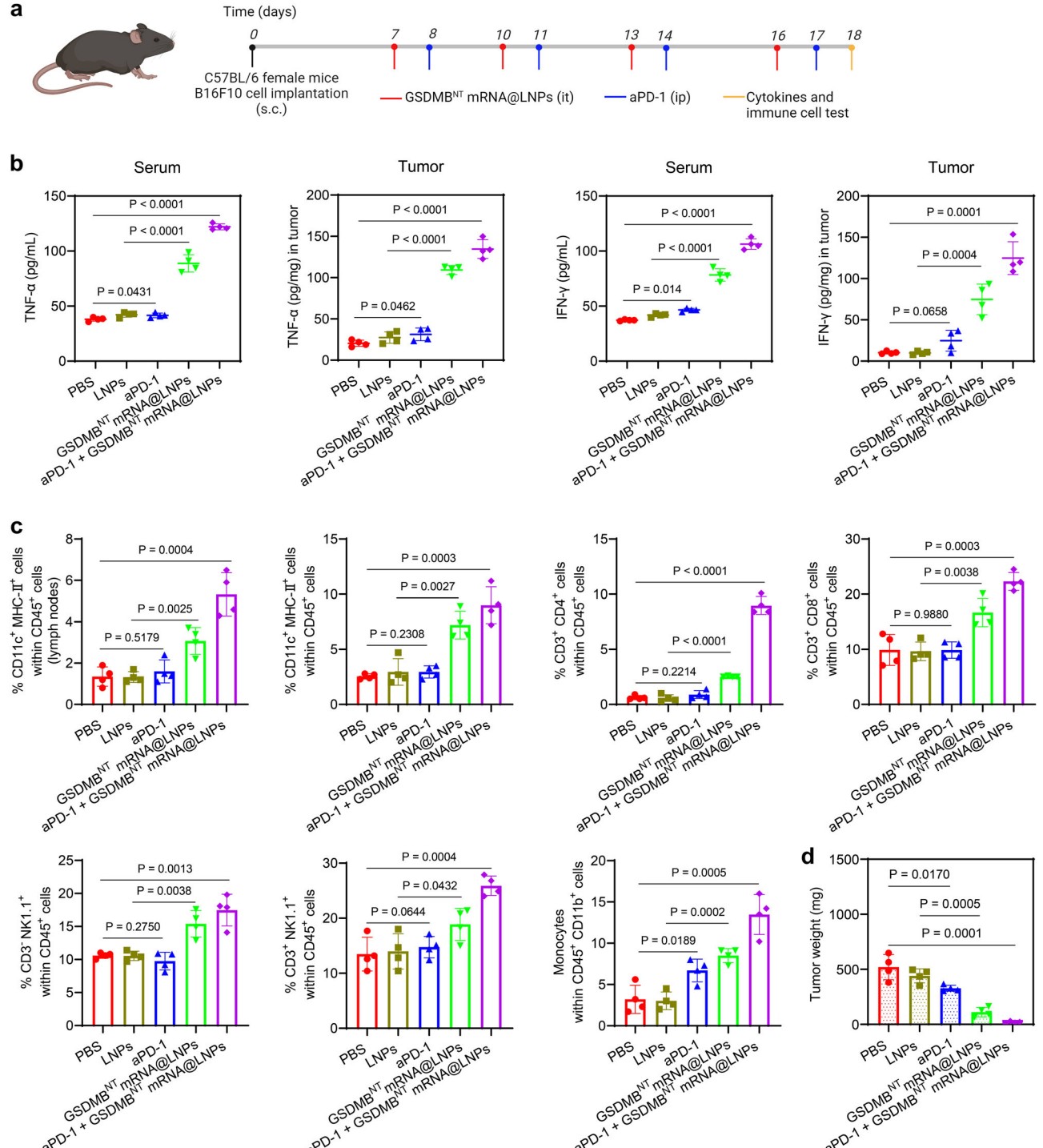

**Fig. 7 | Treatment with GSDMB^NT mRNA@LNPs remodels the tumor microenvironment in an aggressive melanoma mouse model. a** The experimental timeline for the treatment of B16F10 tumor-bearing mice and cytokines and immune cell tests were conducted on day 18. **b** Cytokine concentrations were measured in tumor tissue or serum by ELISA. **c** Flow cytometry analysis results of the percentage of CD11c⁺ MHC-II⁺ DCs, CD3⁺ CD4⁺ T cells, CD3⁺ CD8⁺ T cells, CD3⁻

NK1.1⁺ T cells, CD3⁺ NK1.1⁺ T cells, and monocytes isolated from lymph nodes or tumors. **d** Tumor weights of B16F10 tumor-bearing mice with different treatments. All results are presented as means ± SD (*n* = 4 mice per group). Statistical significance was calculated using a two-tailed Student's *t* test. Source data are provided as a Source Data file. Panel **a** was created with BioRender.com.

cancer immunotherapy, as it has successfully been shown to inhibit checkpoint proteins to augment the host's immunologic activity against tumors[4]. However, resistance to immunotherapy occurs frequently due to immunosuppressive cytokines and insufficient tumor-infiltrating immune cells in the TME, driving researchers to explore approaches to sensitize immunologically cold tumors. Here, we report

a general mRNA nanomedicine approach showing that single-agent mRNA/LNPs encoding GSDM N-terminal domain trigger inflammatory pyroptosis to turn cold tumors hot. The single-agent pyroptosis-triggering mRNA/LNPs enable robust antitumor immunity and reinforce aPD-1-mediated immunotherapy through reprograming the TME from immunosuppressive into immunostimulatory phenotype.

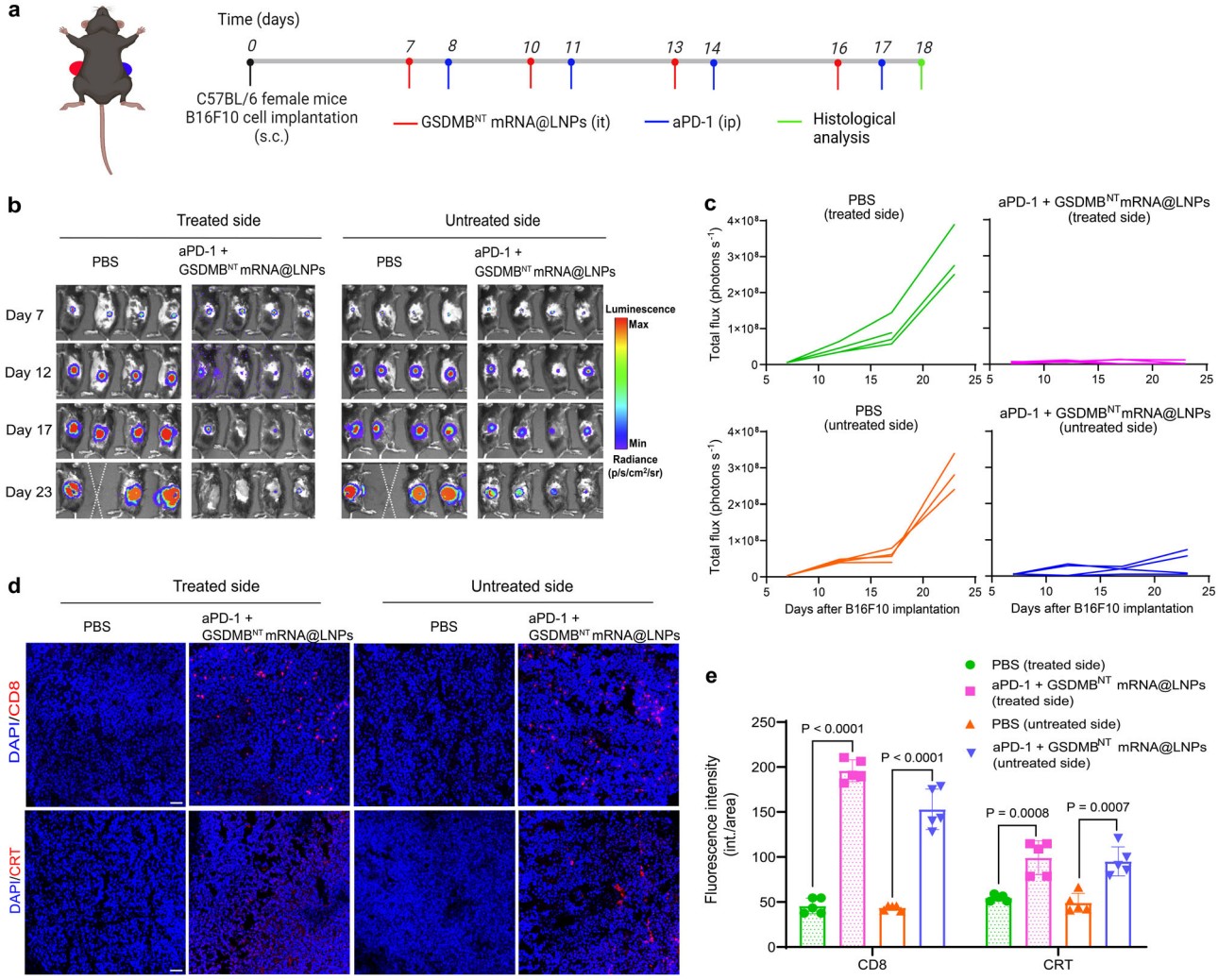

**Fig. 8 | Local combinational treatment of GSDMB^NT mRNA@LNPs and aPD-1 controls tumor burden at distant sites.** C57BL/6 mice (*n* = 4 mice per group) were inoculated subcutaneously (s.c.) with $5 \times 10^5$ and $2.5 \times 10^5$ B16F10 cells in the left and right flanks, respectively. **a** Experimental timeline for treatment of B16F10 dual-tumor-bearing mice. **b** In vivo bioluminescence images of luciferase signals in mice treated as indicated for monitoring tumor growth. Imaging was performed every 5 days from the initial treatment day (day 7 after tumor inoculation) until day 23.

**c** Individual curves of luciferase signals for mice treated as indicated.
**d**, **e** Representative images and quantitative analysis of immunofluorescence staining for CD8+ T cell infiltration and CRT expression in tumors after indicated treatments. Scale bar = 50 μm. Results are presented as means ± SD (*n* = 5). Statistical significance was calculated using a two-tailed Student's *t* test. Source data are provided as a Source Data file. Panel **a** is created with BioRender.com.

GSDM-mediated pyroptosis, a form of cell death accompanied by the secretion of multiple inflammatory cytokines, has recently attracted considerable attention as a unique mechanism in cancer immunotherapy[19–21]. Evidence suggests that the expression of GSDM proteins is suppressed in many cancers[19, 32]. For example, GSDMB is a member of the GSDM protein family that appears to be silenced in gastric and esophageal cancers[20]. It has been shown that the N-terminal domain of GSDMB triggered more pronounced pyroptosis when compared to the corresponding N-terminal domains of other gasdermin proteins[33]. Thus, exogenous GSDMB may potentially act as a tumor suppressor by activating pyroptosis to achieve antitumor immunity. The pyroptosis-triggering mRNA/LNPs presented in this work exert antitumor effects through dual mechanisms. The mRNA nanotherapeutics not only induce direct cell death in cells, but also promote antitumor immunity by inducing ICD, triggering proinflammatory cytokine release, and activating and recruiting immune cells in the tumor microenvironment. Our results imply that this strategy creates a positive feedback loop that converts immunologic cold tumors to hot and establishes a TME that is conducive to sensitizing the tumors to a-PD1 immunotherapy.

Moreover, we found that even 20% cellular pyroptosis triggered by single-agent GSDMB^NT mRNA@LNPs was adequate to promote robust immunogenic cell death, characterized by morphological changes, plasma membrane rupture, cell death, and DAMP secretion. In the 4T1 and B16F10 mouse tumor models, our results further suggest that low levels of cancer cell pyroptosis activated the secretion of proinflammatory cytokines (such as TNF-α, IFN-γ, IL-1β, and IL-18), upregulated the expression of DAMPs, recruited the infiltration of various immune cells, and inhibited the growth of established tumors, thereby establishing a positive feedback loop to promote antitumor immunity. These results were consistent with recent research that showed pyroptosis of less than 15% of tumor cells was sufficient to control over 4T1 mammary tumor graft[21]. In comparison with other strategies in cancer immunotherapy that induce ICD, such as chemotherapy or thermotherapy, GSDM-mediated pyroptosis does not require the killing of large numbers of tumor cells to induce a robust immune response. In a clinical setting, this may be highly advantageous, due to reduced high-dose toxicity and avoiding side effects on normal tissues.

However, conventional GSDMB-mediated pyroptosis requires the caspase or granzyme A-dependent cleavage to free GSDMB[NT] in GSDMB-expressing cells[20]. In our design utilizing GSDMB[NT]-encoded mRNA/LNPs, we are able to deliver free GSDMB[NT] into tumor tissue directly and trigger pyroptosis in a facile and highly efficient manner. Indeed, our results demonstrate that only GSDMB[NT] mRNA@LNPs, but not mRNA@LNPs encoding the full length or C-terminal of GSDMB, induced extensive pyroptosis in transfected cells. When compared with the conventional protein delivery strategy to cleave full-length GSDM proteins, our approach shows that the direct delivery of N-terminus GSDM mRNA/LNPs could trigger pyroptosis simply, immediately, and effectively, which is encouraging for its clinical translation.

T-cell recruitment is a critical challenge for immunotherapy in cold tumors. Our GSDMB[NT] mRNA@LNP treatment recruited tumor-infiltrating CD4[+] and CD8[+] T cells, probably because the treatment initiates or reinitiates a self-sustaining cycle of cancer immunity after intratumoral TNF-α, IFN-γ, IL-1β, and IL-18 induction[34,35]. We also found higher expression of ICD biomarkers in mice treated with pyroptosis-triggering mRNA/LNPs, especially in mice with the combinatorial treatment of GSDMB[NT] mRNA@LNPs and aPD-1. These results reveal that pyroptosis-triggering mRNA/LNPs remodeled the immunosuppressive TME that promotes response to aPD-1. As expected, GSDMB[NT] mRNA@LNPs improved the therapeutic benefits of aPD-1 immunotherapy, including prolonged survival, and enhanced inhibition of tumor growth. In addition, we discovered that a local immune response stimulated by GSDMB[NT] mRNA@LNP treatment in one lesion provoked a systemic antitumor response, contributing to control over distant untreated lesions. These results highlight the possibility to potentiate ICB-mediated immunotherapy in the clinic by synergy with N-terminus GSDM-mediated pyroptosis.

For clinical translation, one additional advantage of our pyroptosis-triggering mRNA/LNP approach is the mRNA/LNP delivery system. mRNA nanomedicine-based gene therapy is a promising therapeutic modality for the treatment of various diseases, due to its excellent safety, quick manufacturing and production, and ability to encode proteins or gene-editing components such as cas9 protein[36–39]. Recently, the two highly effective COVID-19 mRNA vaccines produced by Moderna and Pfizer-BioNTech highlight the enormous potential of mRNA/LNP technology in revolutionizing life science and medical research[40–43]. Although such technological progress justifies more preclinical and clinical studies of mRNA therapeutics, as of now no mRNA nanomedicines have been approved for cancer treatment in clinics[44,45]. The synergistic combination of mRNA encoding inflammatory cytokines or immune agonists with immunotherapy has been extensively explored for enhancing cancer immunotherapy. There have been instances where multiple mRNAs encoding different cytokines were employed for combinatorial cancer therapy. For example, an mRNA mixture of four cytokines, including interleukin-12 (IL-12) single chain, interferon-α (IFN-α), granulocyte-macrophage colony-stimulating factor, and IL-15 sushi, was able to induce a proinflammatory TME and boost antitumor T cell activity, whereas any single cytokine treatment failed to drive effective growth inhibition of established immunologically cold mouse tumors[36]. However, this type of approach usually requires complex manufacturing processes, complicated quality control measures and difficulties also emerge in the analysis of each variable on the clinical outcome. Therefore, a simple and robust single-agent mRNA therapeutic approach presents a promising and favorable solution for enhanced cancer immunotherapy. To this end, our synergistic combination of pyroptosis-triggering mRNA/LNPs with checkpoint immunotherapy provides insights into developing single-agent mRNA nanomedicines for cancer treatment in future clinical practice.

Although encouraging results were achieved here, intratumoral administration may not be applied directly to some solid-organ tumors, such as orthotopic liver or lung tumors. Further explorations in developing organ-targeted or cell-targeted lipid nanoparticles would help broaden the application of pyroptosis-triggering mRNA nanomedicines in various cancers. Additional investigations in T cell-based immunotherapy (e.g., chimeric antigen receptor (CAR) T-cell therapy) and cancer vaccines (e.g., personalized neoantigen vaccine) using the same methods described here would also contribute to further validate the efficacy of the pyroptosis-triggering mRNA nanomedicine platform for potential translation. Dose and frequency also impact the therapeutic benefit, especially since conventional mRNAs require multiple doses, newer developments such as circular mRNA/self-amplifying mRNA may improve the dosing regimen. Therefore, our future work will focus on developing cutting-edge circular mRNA (which displays high levels of protein expression) or self-amplifying mRNA (capable of long-term protein expression) and incorporating them into the nanomedicine platform.

In conclusion, we developed a unique single-agent mRNA nanomedicine that takes advantage of the discovered GSDM-mediated pyroptosis pathway and LNP-meditated mRNA delivery system to improve cancer immunotherapy. Our approach possesses many beneficial properties: (1) GSDMB[NT] mRNA@LNPs not only kill cancer cells directly but also elicit a robust and safe antitumor immunity using a single-agent mRNA; (2) GSDMB[NT] mRNA@LNPs deliver the N-terminus of gasdermin to trigger rapid and efficacious pyroptosis without the need for protease cleavage; (3) GSDMB[NT] mRNA@LNPs reverse the immunosuppressive TME and recruit tumor-infiltrating immune cells, turning cold tumors hot; (4) GSDMB[NT] mRNA@LNPs could be synergized with immune checkpoint blockades to strengthen immunotherapy efficacy, resulting in long-term overall survival, elimination of treated tumors, and stabilization of distant lesions; (5) GSDMB[NT] mRNA@LNPs sensitize aPD-1-mediated immunotherapy in a general manner, thereby they may serve as a basic universal platform to potentially enhance other immunotherapy modalities, such as T cell-based therapies and cancer vaccines. Overall, our single-agent pyroptosis-triggering mRNA nanomedicine therapy is simple and highly efficacious, displaying great potential for clinical translation.

## Methods

### Preparation and characterization of mRNA-encapsulating LNPs

The sequences for in vitro transcription of mRNA, including T7 promoter[46,47], 5′ UTR, coding sequence, 3′ UTR, and poly(A) (Supplementary Table 1) were cloned into pVAX1 vector using NEBuilder® HiFi DNA Assembly Cloning Kit. The mixture was then transformed into DH5α Competent Cells (ThermoFisher) by chemical transformation and the synthesized plasmid was confirmed by Sanger Sequencing. Then, a linearized DNA template, including T7 promoter, 5′ UTR, coding sequence, 3′ UTR, and poly(A) was achieved by BsaI digestion. The DNA templates were purified by QIAquick Gel Extraction Kit (Qiagen) and confirmed by 1% agarose gel electrophoresis. All mRNAs were synthesized by in vitro transcription with ARCA (TriLink) or CleanCap (TriLink) and 100% pseudouridine-5′- triphosphate (APEx-BIO) using AmpliScribe T7-Flash Transcription Kit (Lucigen) following the manufacturer's instruction. Subsequently, mRNA was purified by RNA Clean & Concentrator (Zymo). The synthesized mRNAs were examined by 1% agarose gel electrophoresis and stored at -80 °C for future use.

LNP formulations were prepared as previously described[26]. Briefly, lipids were dissolved in ethanol at molar ratios of 40:40:25:0.5 (AA3-DLin: DOPE: cholesterol: PEG-2000). The lipid mixture was combined with a 25 mM Sodium acetate buffer solution (pH 5.5) containing mRNA at a ratio of 20:1 (AA3-DLin: mRNA, wt./wt.) for in vitro study and 10:1 for in vivo study. Formulations were dialyzed against PBS (pH 7.4) in dialysis cassettes overnight.

The hydrodynamic diameters and zeta potentials of mRNA-encapsulating LNPs were analyzed by dynamic light scattering (DLS)

on a Malvern Instruments Zetasizer HS III (Malvern, UK) at room temperature. Transmission electron microscopy (JEM-F200 TEM, USA) was performed to detect the morphology of mRNA-encapsulating LNPs. To investigate the stability, the size changes of mRNA-encapsulating LNPs were measured in PBS, medium, medium with 10% FBS, pH = 7.4 buffer, pH = 6.5 buffer, 10% plasma or 20% plasma at 0, 2, 4, 6, 8, 24, 48, and 72 h.

### Cell culture and transfection

HEK 293, HeLa, 4T1, and B16F10-Luc cell lines were obtained from the American Type Culture Collection (ATCC). All cells were cultured in Dulbecco's modified Eagle's medium (DMEM) containing 10% FBS, 100 units/mL penicillin, and 100 mg/mL streptomycin. Cells were grown in a humidified atmosphere with 5% $CO_2$ at 37 °C.

For in vitro transfection, 1 µg mRNA was encapsulated in LNPs as described above. The formulated mixtures were added into a well of a 12-well plate containing 1.0 mL medium. At predetermined time points after transfection, the cells or supernatants were collected for further assays. For in vivo transfection, 10 µg mRNA was formulated as above and intratumorally administered for various assays.

### Western blot analysis

HEK 293 cells were seeded in a six-well plate at a density of $5 \times 10^5$ cells per well. After cells reached 70–80% confluence, cells were treated with naked GSDMB[NT] mRNA or GSDMB[NT] mRNA@LNPs. Untreated cells served as a control (Ctrl). Protein samples were extracted from cells lysed with radioimmunoprecipitation (RIPA) buffer supplemented with protease inhibitors. Protein samples were separated by sodium dodecyl-sulfate polyacrylamide gel electrophoresis (SDS-PAGE), transferred to polyvinylidene fluoride (PVDF) membranes and then blocked with 5% milk. Diluted primary GSDMB antibody (Supplementary Table 2) was incubated with the membranes overnight, followed by incubation with secondary antibody for 2 h at 37 °C. After washing three times with TBST (20 mM Tris, 160 mM NaCl, 0.1% Tween 20), the ChemiDoc XRS system (Bio-Rad) was used to detect the chemiluminescent signals.

### Cellular uptake activity and endosomal escape of mRNA@LNPs

HEK 293, HeLa, 4T1, and B16F10-Luc cells were seeded on glass slides overnight, followed by transfection with Luc mRNA@[FITC]LNPs for 0.5, 2, or 4 h. The fluorescence images were acquired with a Nikon A1R + HD Confocal Microscope. For the endosomal escape study, cells were seeded in confocal dishes at a density of $1 \times 10^5$ cells. Then, cells were transfected with LNPs encapsulating 0.5 µg/mL Luc [Cy5]mRNA. After 4 h, cells were stained with DAPI (ThermoFisher) and LysoBrite™ Green (AAT Bioquest), and the fluorescence signals of these cells were analyzed by CLSM.

### In vitro evaluation of pyroptosis

To examine the changes in cell morphology after GSDMB[NT] mRNA@LNPs transfection, HEK 293, HeLa, 4T1, and B16F10-Luc cells were seeded in 35 mm Petri dishes containing 1.0 mL DMEM medium. After cell attachment, cells were treated with naked GSDMB[NT] mRNA, LNPs, or GSDMB[NT] mRNA@LNPs, respectively. After 24 h of treatment, annexin V-FITC and propidium iodide were added to the cell culture medium. After incubation for 15 min in the dark, a Nikon A1R + HD Confocal Microscope was used to capture live cell images for studying changes in cell morphology. The images shown are representative of at least three randomly selected fields. To quantitatively analyze pyroptotic cells, flow cytometry was performed to determine the number of annexin V-FITC and PI-positive cells. All cells collected from each 12-well plate were washed twice with Annexin V binding buffer and stained by using a FITC Annexin V Apoptosis detection kit with PI (BioLegend). The release of LDH was measured with a CyQUANT™ LDH-Cytotoxicity Assay Kit, following the manufacturer's instructions. Calcein acetoxymethyl (Calcein-AM) release assay was used to determine cell death

resulting from pyroptosis-induced cell lysis. Briefly, 2% Triton X-100 was added to cells in the control group for 2 h, which was used to determine the maximum release by nonspecific lysis. Cells without any treatment were used to determine the spontaneous release of calcein. Subsequently, 4 µg/mL calcein AM was added to the cell culture medium for 30 min at 37 °C. The fluorescence intensity of the released calcein was measured under an excitation wavelength of 485 nm and an emission wavelength of 530 nm. The calcein release was calculated according to the formula: [(*test release -spontaneous release*)/(*maximum release by nonspecific lysis - spontaneous release*)].

### In vitro stimulation of DCs

BMDCs were isolated from the bone marrow of female C57BL/6 mice as described previously[48]. Briefly, single-cell suspensions were harvested from femurs and tibias, passed through 70-µm nylon cell strainers and then cultured in RPMI 1640 complete medium with 20 ng/mL M-CSF (Peprotech) for one week, in preparation for future experiments. B16F10 cells were pretreated with PBS (Ctrl), naked GSDMB[NT] mRNA, blank LNPs or GSDMB[NT] mRNA@LNPs, followed by coculture with BMDCs for 48 h. An ELISA assay was also performed to evaluate the production of pro-inflammatory cytokines (IFN-γ, IL-1β and TNF-α) in the medium. The maturation biomarkers for DCs (MHC-II and CD86) were measured by flow cytometry.

### In vitro cytotoxicity on macrophages and DCs

Macrophages and DCs were isolated from the bone marrow of C57BL/6 mice and were seeded in 96-well plates at a density of $5 \times 10^4$ cells per well. Then, cells were treated with GSDMB[NT] mRNA@LNPs at different mRNA concentrations (0.25, 0.50, 1.00, 1.25, and 1.5 µg/mL). After 48 h of incubation, 10 µL of CCK-8 was added to each well and incubated for a further 3 h. The absorbance at 450 nm for each sample was measured on a microplate reader to calculate cell viability (%).

### Cell immunofluorescent staining

HEK 293, HeLa, 4T1, and B16F10-Luc cells were seeded on glass slides in 12-well plates and were incubated with naked GSDMB[NT] mRNA, LNPs, or GSDMB[NT] mRNA@LNPs, respectively. After 48 h, cells were washed twice with PBS and fixed with 4% paraformaldehyde for about 20 min at room temperature. Then, the cells were permeabilized with 0.1% Triton X-100 PBS (PBST) and blocked with a PBST blocking solution containing 5% goat serum for 1 h at room temperature. Next, the cells were incubated in the diluted CD8 or CRT antibody (Supplementary Table 2) in PBST in a humidified chamber overnight at 4 °C. After that, cells were washed with PBS three times and incubated in secondary antibody with Alexa Fluor 594 for 1 hour at room temperature. Then, cells were washed with PBS three times and stained with 4′,6-diamidino-2-phenylindole (DAPI) (Thermo Fisher Scientific) for 10 min to visualize cells nuclei. Finally, cells were sealed with a drop of mounting medium and analyzed by a Nikon A1R + HD Confocal Microscope at a wavelength of 594 nm. The images shown are representative of at least three randomly selected fields. For CRT exposure analysis, the cells were collected after treatment and stained with an Alexa Fluor 488-labeled CRT antibody (Supplementary Table 2) for 1 hour at 4 °C, washed with PBS three times, and stained with PI. Then, samples were analyzed by flow cytometry (BD Biosciences, San Jose, CA). All flow cytometry data were analyzed using FlowJo software.

### Animals and mouse tumor models

All animal studies were approved by the Institutional Animal Care and Use Committee (IACUC) at Rutgers University. Balb/c female mice and C57BL/6 female mice aged 6−8 weeks were purchased from the Jackson Laboratory and housed in a temperature-controlled environment on a 12-h light cycle with free access to food and sterile water. All mice were allowed to acclimate for at least 3 days before tumor cell implantation. For an orthotopic breast tumor model,

$5 \times 10^5$ 4T1 cells in 50 µL of sterile PBS were injected into the fat pad of the fourth pair of the left breast of Balb/c female mice. To establish a B16F10-bearing mouse model, B16F10-Luc cells ($5 \times 10^5$) in 100 µL of sterile PBS were implanted subcutaneously into the right flank of C57BL/6 female mice. For the B16F10 dual-flank tumor model, $5 \times 10^5$ cells were implanted subcutaneously on the left side, and $2.5 \times 10^5$ cells were implanted subcutaneously on the right side on the same day. Regarding the in vivo rechallenge experiment, B16F10 tumor-bearing mice that had previously received a combination treatment regimen of aPD-1 and GSDMB$^{NT}$ mRNA@LNPs were rechallenged with $5 \times 10^5$ B16F10 cells on the left flank. Naive mice were subcutaneously implanted with the same number of B16F10 cells on day 0 to serve as a control. Tumor volume was measured every two days using a Vernier caliper and calculated as $V = (a \times b^2)/2$, where $a$ is the long axis and $b$ is the short axis. In this study, the tumor burden did not exceed the permitted diameter of 2 cm as outlined by the ethics guidelines of Rutgers University's IACUC.

### In vivo imaging and drug administration

Luciferase signals were analyzed by an in vivo imaging system (IVIS, PerkinElmer) to examine the in vivo translation efficiency of Luc mRNA and monitor the growth of B16F10-Luc-bearing tumor models. Briefly, 100 µL IVISbrite d-Luciferin potassium salt bioluminescent substrate (15 mg/mL in PBS) was injected intraperitoneally. After 10 min, mice were imaged in the imaging system. Luminescence intensity was quantified using the living image software (PerkinElmer).

For tumor therapy, one week after tumor inoculation, mice were received with PBS, LNPs, GSDMB$^{NT}$ mRNA@LNPs, or aPD-1 as indicated. Briefly, 50 µL of GSDMB$^{NT}$ mRNA@LNPs was administered intratumorally into the tumors on days 7, 10, 13, and 16 for a total of four doses. Anti-mouse PD-1 antibodies (Bio X Cell, clone RMP1-14) were injected intraperitoneally at a dose of 100 µg on days 8, 11, 14, and 17 for a total of four doses. During the study, mice were checked daily for adverse clinical reactions. The body weight of mice was monitored every two days until the end of the experiments.

### ATP, HMGB1, and cytokines detection

HEK 293, HeLa, 4T1, and B16F10-Luc cells were seeded in 12-well plates and were incubated with GSDMB$^{NT}$ mRNA, LNPs, or GSDMB$^{NT}$ mRNA@LNPs. After 48 h, the cell culture medium was collected for extracellular ATP assay (Promega) and extracellular HMGB1 analysis (Chondrex).

To evaluate intratumoral HMGB1 and cytokines including TNF-α, IFN-γ, IL-1β, and IL-18, tumor tissues were excised and homogenized in tissue extraction reagents including 1% proteinase and phosphatase inhibitors (Thermo Fisher Scientific). The supernatant from tumor homogenates was then measured with ELISA kits (cytokine kits purchased from Thermo Fisher Scientific).

### Flow cytometry for immune cells and antibodies

Antibodies for flow cytometry analysis are listed in Supplementary Table 2. Purified anti-mouse CD16/32 antibody (clone 93), PerCP/Cyanine5.5 anti-mouse CD45.2 Antibody (clone 104), APC anti-mouse CD11c Antibody (clone N418), FITC anti-mouse I-A/I-E Antibody (clone M5/114.15.2), APC anti-mouse CD3 Antibody (clone 17A2), FITC anti-mouse CD4 Antibody (clone GK1.5), PE anti-mouse CD8a Antibody (clone 53-6.7), FITC anti-mouse NK-1.1 Antibody (clone PK136), FITC anti-mouse/human CD11b Antibody (clone M1/70), APC anti-mouse Ly-6C Antibody (clone HK1.4) and PE/Cyanine7 anti-mouse Ly-6G Antibody (clone 1A8) were purchased from BioLegend.

The expression of stimulatory markers of DCs (CD11c$^+$ and major histocompatibility complex II$^+$ (MHCII$^+$)), natural killer (NK) cells (CD3$^-$NK1.1$^+$), NK T cells (CD3$^+$NK1.1$^+$), and monocytes (CD11b$^+$Ly6g$^-$Ly6c$^+$) were analyzed by fluorescence-activated single cell sorting (FACS). Briefly, tumors and lymph nodes were harvested and digested by 1 mg/mL collagenase IV (Thermo Fisher Scientific) for 30 min at 37 °C to make single-cell suspensions. The single-cell suspensions were then passed through 70-µm nylon cell strainers. The suspension was centrifuged, and the cell pellets were washed and resuspended in the PBS containing 1% BSA (FACS buffer), blocked with anti-mouse CD16/CD32 for 30 min, and finally stained with the indicated antibodies for another 1 h. Stained samples were analyzed using a FACS analyzer (BD Biosciences, San Jose, CA). All flow cytometry data were analyzed using FlowJo software.

### Immunofluorescence, histopathology, and in vivo evaluation of pyroptosis

At the end point of treatment, the tumors and organs were harvested and embedded in OCT tissue cassettes and frozen on dry ice for sectioned into slices at a thickness of 10 µm. For tissue immunofluorescence, sample sections were fixed with 4% paraformaldehyde for 30 min and then permeabilized with 1% Triton X-100 PBS (PBST) and blocked with a PBST blocking solution containing 5% goat serum for 1 hour at room temperature. Next, the sample slides were incubated with diluted CD8 or CRT antibody (Supplementary Table 2) and imaged as described in the "cell immunofluorescence" section. For histopathology analysis, tissue sections with 10 µm thickness were stained with hematoxylin and eosin (H&E) for pathology following the manufacturer's instructions.

To further evaluate the tumor cell pyroptosis in vivo, mice were intravenously treated with 2.5 mg/kg propidium iodide. After 20 min, tumors and major organs were harvested and then embedded into OCT-containing Cryomold molds and frozen for sectioned into slices at a thickness of 10 µm. After slicing and mounting, the tissue sections were imaged directly and immediately on a fluorescence microscope (BZ-X710; Keyence, Kyoto, Japan).

To investigate the cell death rate in vivo, we harvested tumor tissues from B16F10 tumor-bearing mice that received either PBS or a combination treatment of GSDMB$^{NT}$ mRNA@LNPs and a-PD1, in order to evaluate the population of PI-positive cells using flow cytometry analysis.

### In vivo safety evaluation

To evaluate the in vivo safety profile of GSDMB$^{NT}$ mRNA@LNPs, the body weight of mice was monitored every two days until the end of the experiments. On the second day after the final injection, tumor tissues and retro-orbital blood samples were collected for histological and aminotransaminase analyses.

### Statistical analysis

All results are analyzed using GraphPad Prism software and presented as the means ± SD. Unpaired t-test and one-way ANOVA were used for two-group or multiple-group comparisons. The details of statistical analysis for figures and Supplementary Figures are performed as indicated in the figure legends, and survival analysis was analyzed using the log-rank test.

### Reporting summary

Further information on research design is available in the Nature Portfolio Reporting Summary linked to this article.

## Data availability

All data generated or analyzed during this study are included in the Article and its Supplementary Information file and the Source Data file. Source data are provided with this paper.

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

## Acknowledgements

X.X. acknowledges funding from National Science Foundation (2001606) and American Heart Association grant #19AIREA34380849.

This research is also supported by the Gustavus and Louise Pfeiffer Research Foundation Award. X.-Q. Z. acknowledges funding from the "Open Competition to Select the Best Candidates" Key Technology Program for Nucleic Acid Drugs of NCTIB (Grant No. NCTIB2022HS02002), the Natural Science Foundation of Shanghai (23ZR1427600), Shanghai Jiao Tong University Scientific and Technological Innovation Funds (2019TPA10), Foundation of National Facility for Translational Medicine (Shanghai) (TMSK-2020-008), and the Interdisciplinary Program of Shanghai Jiao Tong University [project number ZH2018ZDA36 (19×190020006)].

## Author contributions

F.L. and X.-Q.Z. contributed equally. X.X., F.L., and X.-Q.Z. conceived the project and designed the experiments. F.L. performed most experiments. F.L., X.-Q.Z. and X.X. wrote the manuscript. F.L., W.H., M.T., Z.L., L.B., X.-Q.Z. and X.X. discussed the results and reviewed the conclusions. W.H. helped with the in vivo experiments. Z.L. assisted with LNP fabrication. X.X., X.-Q.Z., F.L., W.H., and M.T. reviewed and edited the manuscript. F.L., X.-Q.Z., and X.X. were mainly responsible for data interpretation.

## Competing interests

The authors declare that they have no competing interests.
