## [Peer Review File · Nature Communications]

mRNA lipid nanoparticle-mediated pyroptosis sensitizes immunologically cold tumors to checkpoint immunotherapyREVIEWER COMMENTS

Reviewer #1 (Remarks to the Author): with expertise in cancer nanotherapy

In this manuscript, the authors developed lipid nanoparticles to achieve mRNA transfection for the generation of the N-terminus of gasdermin and subsequent pyroptosis in tumor cells. The nanoparticles successfully achieved pyroptosis and increased the CRT exposure and HMGB1 release in different tumor cells. Moreover, the therapeutic strategy potently activated the immune system and demonstrated encouraging anti-tumor efficacy in the 4T1 breast cancer mouse model and the B16F10 melanoma model. Overall, the topic is interesting, the data support the conclusions well, and the manuscript shows clear and rigorous logic. Therefore, I suggest the acceptance of this manuscript after addressing the following issues.

1. For the verification of the pyroptosis in vitro, the LDH release assay after the treatments should be investigated to further substantiate the anti-tumor efficacy in vitro.
2. For mRNA delivery or gene delivery, transfection efficacy is important. Thus, the transfection efficacy of the nanoparticles in different kinds of cells is suggested to be calculated in this manuscript.
3. The in vitro cell uptake of the nanoparticles is suggested to be investigated by confocal assay or flow cytometry assay.
4. Typically, the acidic environment and hydrolytic enzymes in the lysosome can degrade the carrier and the mRNA inside. Therefore, endosomal escape before mRNA degradation is considered a critical step for the success of mRNA therapy and should be supplemented or explained.

Reviewer #2 (Remarks to the Author): with expertise in cancer nanotherapy

In this manuscript, the authors reported a unique strategy to reprogram tumor immune microenvironment by using mRNA nanoparticles to restore gasdermin B (GSDMB) N-terminal expression in tumor cells. The GSDMB N-terminal domain can induce pore formation on cell membrane, triggering the immunogenic cell death by releasing of DAMPs and cytokines. These mRNA nanoparticles were well characterized in vitro and in vivo. The therapeutic efficacy of GSDMB(NT) mRNA NPs was systematically examined in multiple tumor models in combination with anti-PD1 immune checkpoint blocker, as well as the change of tumor immune microenvironment. Overall, this work provides a new promising strategy for turning 'cold' tumors 'hot'. Below are several concerns that the authors need to further address.

1. Several gasdermin family proteins can induce pore formation. The choice of gasdermin B in this work needs to be discussed.
2. The GSDMB(NT) mRNA NPs could also be taken up by immune cells (e.g., macrophages and DCs) in the tumors after i.t. injection. Will this induce immune cell toxicity and how could this impact the immune microenvironment change?
3. The anti-tumor effect of GSDMB(NT) mRNA NPs is attributable to the anti-tumor immune response, or the intrinsic pyroptosis/tumor suppressor effect of GSDMB, or both?
4. The release of proinflammatory cytokines into blood might raise some concern on systemic safety.
5. In the study with distant tumors, the authors need to explain what induced CRT expression in the untreated tumors and may further measure the GSDMB(NT) expression in them.
6. The authors claimed that "However, even mRNAs encoding multiple cytokines still failed to induce successful antitumor immunity". The authors should be careful with the claim and at least cite references to support that.

Reviewer #3 (Remarks to the Author): with expertise in cancer immunology, immunogenic cell death

In the manuscript, Li et al. developed an elegant approach to trigger pyroptosis by mRNA lipid nanoparticle in order to sensitize immunologically “cold” tumors to checkpoint immunotherapy. The authors developed mRNA lipid nanoparticles (LNPs) encoding only the N-terminus of gasdermin to trigger pyroptosis. In the next part of the work, several mouse models have been used to analyze anti-tumor immunity. Additionally, the authors demonstrate that mRNA-mediated pyroptosis sensitized tumors to anti-PD-1 immunotherapy.

This is an interesting and clinically relevant approach. However, I strongly believe that in its current version, the manuscript is not mature enough to be published in Nature Communications. However, I suggest several key experiments which will help the authors to generate a revised version with justified and straightforward conclusions.

Major comments

1. Annexin V/propidium iodide (PI) staining is not specific for apoptosis detection (Fig. 2). It is a general cell death detection technique that can identify the stage of cell death: AnV+PI⁻: early-stage and AnV+PI⁺ late stages. It is very well known that PS exposure can occur also in other cell death modalities such as necroptosis. Therefore, it is not a specific cell death marker/technique. The authors should also correct the terminology accordingly and it is necessary to show data on the early and late stages of cell death.
2. The authors should provide clear experimental evidence that GSDMBNT mRNA@LNP induces pyroptosis (and not for example apoptosis and especially these should be shown for 4T1 and B16F10 cells) because in the current version of the manuscript, this experimental evidence is lacking. Moreover, it is necessary to discriminate this cell death induced by GSDMBNT mRNA@LNP from apoptosis and necroptosis.
3. What about the IL-1beta release? Can it be detected in the SN of dead/dying cells? This should be experimentally demonstrated.
4. Surface-exposed calreticulin (CRT) was measured by confocal microscopy (Fig. 3). The description of how it was done is lacking in the Methods section. Of note that CRT exposure can be done only on flow cytometry and it should be analyzed on PI⁻ cell population. Now, the authors demonstrate unspecific intracellular staining, and this could not be interpreted as surface exposure.
5. Analysis of the release of ATP and HMGB1 should be done in parallel with cell death analysis (See Fig. 3). Sometimes ATP can be released premortem before the appearance of the ruptured plasma membrane.
6. Since the authors claim that this is pyroptosis IL-1b and IL-18 should be also analyzed in the tumor and serum of the mice experiments (see Fig. 4).
7. Again, the authors used confocal microscopy to demonstrate CRT exposure at the cell surface (Fig 5). It is not specific because it does not discriminate from the intracellular staining. Another approach should be used to confirm this conclusion.

The same is true for HMGB1 which the authors analyzed by WB. It shows just an increase in expression and has nothing to do with the extracellular release. Please use other techniques to demonstrate HMGB1 release in vivo. For example, one can measure it in the blood/serum.

8. From the presented data it is not clear whether GSDMBNT mRNA@LNP-induced pyroptosis can be actually induced in vivo. It is necessary to demonstrate the cell death rate and type in vivo for 4T1

and B16F10 cellular models.

9. The authors should show that antigens derived from pyroptotic cells can be presented by APC and can induce specific CD8 T cell responses. One can use OVA and OT-1 and OT-2 transgenic mice models.

10. In addition, it is also necessary to demonstrate the uptake of pyroptotic cells in vitro by bone-marrow-derived dendritic cells and to analyze their maturation/activation (MHC-II, CD80, CD86, production of pro-inflammatory cytokines) of dendritic cells.

11. One of the golden standards to demonstrate that indeed the given cell death type is ICD one should use the tumor prophylactic vaccination model. It is strongly advised to add such data to the revised version of the manuscript.

12. The authors should demonstrate that indeed pyroptosis is important for the sensitization of immunologically “cold” tumors to checkpoint immunotherapy. Adding some important controls (e.g., non-ICD) will re-enforce their conclusions.

Minor comments

A description of the statistical methods should be indicated in each figure legend. The number of experiments should be also indicated in each figure legend. Now it is missing, e.g., see figure legend of Fig. 3.

Reviewer #4 (Remarks to the Author): with expertise in pyroptosis, immunology

Inducing pyroptosis is considered to be a promising strategy of cancer treatment. In this manuscript, the author developed a general mRNA nanomedicine approach to turn the immunosuppressive cold tumors into inflammatory hot tumors. This conversion promotes the infiltration of immune cells in TME, enhancing the antitumor activity of anti-PD-1 immunotherapy. They proved that the pyroptosis triggering-mRNA is effective in many kinds of tumor cell as well as in vivo. In summary, the author discovered a novel pyroptosis triggering therapy, which can also enhance the efficacy of other immunotherapies. However, there are still some key issues needed to be addressed.

Major points

1. It would be better if the author could explain the reason for choosing GSDMB instead of other GSDM family members in this therapy? Is GSDMB-NT the most efficient one to induce pyroptosis? Besides, considering that GSDMB is not expressed in rodents, mouse cells may lack the mechanism of post-translational modification of GSDMB and inhibition of pore-forming activity compared with human cells. Therefore, it is questionable whether it is reasonable to study the function of GSDMB using mice as animal models. In addition, female mice were chosen to perform in vivo experiments in this study. Is there any difference in anti-tumor effect of LNP-GSDMB-NT between male and female mice?

2. As a method of injection therapy of exogenous substances, the stability and persistence of exogenous GSDMB-NT mRNA in cells should be fully considered, because the activity of RNase in many tumor cells is relatively high. Meanwhile, the metabolism of the material, including ionizable cationic lipid (AA3-Dlin), phospholipid (DOPE), cholesterol, and PEG, used for mRNA transport, also need to be considered, so as to illustrate the efficiency and side effects of this strategy in a long-term use.

3. Initially, the authors examined the stability of GSDMB-NT mRNA@LNPs in PBS and cell culture medium. However, tumor microenvironment is a special physiological/pathological environment, and its pH and esterase activity may affect the stability of lipid nanoparticles. Therefore, it is necessary to re-explore GSDMB-NT mRNA@LNPs stability in some typical tumor extreme microenvironments.

4. As human tumors are much larger than those in mouse models, the treatment of GSDMB-NT

mRNA@LNPs injected into human tumor lesions should also require more precisely. Is it better to be near the outside of the tumor, favoring the release of cytokines into the surrounding environment to elicit stronger anti-tumor immunity, or is it better to be inside the solid tumor, acting more thoroughly and uniformly from the inside out across the tumor area. Most importantly, intra-tumoral administrated is widely considered as impractical in clinical and four-dose usage adds troubles for its application. 5. As shown in Fig 3, the LNPs used in the current study are not tumor-specific and also target normal cells, resulting in killing of both targeted tumor and normal cells. Thus the strategy described in this study is not as “safe” as the authors mentioned in the discussion. It is better to modify the LNPs with appropriate groups to improve the specificity of LNPs.

Minor points

1. In figure 2D (corresponding description in line 122-125), what the treatment of GSDMB-NT mRNA group stands for? Was the mRNA added to the cell culture medium directly or transfecting cells with other method? It would be clearer if the author could describe this control group either in figure legend or in the methods part.
2. The authors need to confirm the expression of GSDMB-NT in LNPs targeted cells by immunoblot analysis. Additionally, since the authors mentioned adverse clinical reactions were not observed in the mouse models, these data need to be included in the manuscript.

RESPONSE TO REVIEWER COMMENTS

Reviewer #1 (Remarks to the Author): with expertise in cancer nanotherapy

In this manuscript, the authors developed lipid nanoparticles to achieve mRNA transfection for the generation of the N-terminus of gasdermin and subsequent pyroptosis in tumor cells. The nanoparticles successfully achieved pyroptosis and increased the CRT exposure and HMGB1 release in different tumor cells. Moreover, the therapeutic strategy potentially activated the immune system and demonstrated encouraging anti-tumor efficacy in the 4T1 breast cancer mouse model and the B16F10 melanoma model. Overall, the topic is interesting, the data support the conclusions well, and the manuscript shows clear and rigorous logic. Therefore, I suggest the acceptance of this manuscript after addressing the following issues.

We thank the reviewer's evaluation and the positive feedback. We have addressed the comments by adding additional data below.

1. For the verification of the pyroptosis in vitro, the LDH release assay after the treatments should be investigated to further substantiate the anti-tumor efficacy in vitro.

Response: We would like to express our sincere gratitude to the reviewer for taking the time to read and provide insightful comments on our paper. The feedback has been extremely helpful in improving the quality of our work. As suggested, we conducted an LDH release assay to investigate the lethal effect of GSDMB^{NT} mRNA@LNPs in different types of cells. As shown in **Fig. R1**, GSDMB^{NT} mRNA@LNP treatment led to markedly increased cell death rates of 51.5%, 47.5%, 32.6%, and 23.3% for HEK 293, HeLa, 4T1, and B16F10 cells, respectively, compared to the control groups (naked GSDMB^{NT} mRNA or blank LNPs) which exhibited substantially lower rates of below 5% for all four cell types. We have included the relevant Methods and Results in the revised manuscript (page 7, lines 146-151; page 22, lines 501-502). **Fig. R1** has also been included in **Fig. 2** as **Fig. 2b**. All changes have been highlighted in yellow.

Fig. R1 LDH release-based cell death assay in HEK 293, HeLa, 4T1, and B16F10 cells after treatment with naked GSDMB^{NT} mRNA, LNPs, or GSDMB^{NT} mRNA@LNPs, respectively. Data are presented as means \pm SD (n = 3). Statistical significance was calculated via one-way ANOVA.

2. For mRNA delivery or gene delivery, transfection efficacy is important. Thus, the transfection efficacy of the nanoparticles in different kinds of cells is suggested to be calculated in this manuscript.

Response: We appreciate the reviewer for this comment. In addition to fluorescence imaging analysis, we used flow cytometry to quantitatively determine the percentage of positive cells transfected with mCherry-loaded LNPs in four cell lines as suggested and included the data in the updated **Extended Data Fig. 4** (designated as **Fig. R2** in this letter). As shown in **Fig. R2**, the mCherry-loaded LNPs showed transfection efficacies of 83.5%, 75.6%, 46.8% and 45.3% in HEK 293, HeLa, 4T1 and B16F10 cells, respectively. We have highlighted the updated results on page 7, lines 138-140 in the revised manuscript.

Fig. R2 Transfection efficacy of LNPs was determined by fluorescence microscope (a) and flow cytometry analysis (b) in four different cell lines.

3. The in vitro cell uptake of the nanoparticles is suggested to be investigated by confocal assay or flow cytometry assay.

Response: Thank you for this comment. As suggested, we formulated FITC-labeled LNPs (FITC^LNPs) and evaluated the in vitro cell uptake of the LNPs by confocal microscopy at 0.5, 2, and 4 hours. **Fig. R3** shows that the intracellular 488 nm green signal derived from FITC^LNPs increases proportionally with the incubation time, demonstrating that all the tested cell lines uptake the LNPs efficiently. This result has been included in the updated **Fig.1** as **Fig. 1e**. We

have highlighted the updated Methods and Results on page 6, lines 127-131; pages 22, lines 482-485 in the revised manuscript.

Fig. R3 Cellular uptake of FITC-labeled LNPs (Luc mRNA@^{FITC}LNPs) was monitored in HEK 293, HeLa, 4T1, and B16F10 cells at different time points. Scale bar = 20 μ m.

4. Typically, the acidic environment and hydrolytic enzymes in the lysosome can degrade the carrier and the mRNA inside. Therefore, endosomal escape before mRNA degradation is considered a critical step for the success of mRNA therapy and should be supplemented or explained.

Response: We agree with the reviewer's comments that endosomal escape is crucial for effective mRNA delivery. As suggested, Luc ^{Cy5}mRNA-loaded LNPs (termed Luc ^{Cy5}mRNA@LNPs) were formulated, and their endosomal escape ability was studied. **Fig. R4** (Extended Data **Fig. 3** in the updated Supplementary information) demonstrated that the majority of the red signal from ^{Cy5}mRNA was not colocalized with the green signal from the endosomes or lysosomes after LNP incubation with cells for 4 hours, revealing the successful cytoplasmic release of mRNA by LNPs. We have included the relevant Methods and Results in

the revised manuscript (Page 6, lines 131-135; Page 22, lines 485-489). **Fig. R4** has also been included in **Fig. 1** as **Fig. 1f**. All changes have been highlighted in yellow.

Fig. R4 LNP-mediated endosomal/lysosomal escape and cytoplasmic release of Luc^{Cy5}mRNA in HEK 293, HeLa, 4T1, and B16F10 cells 4 hours after incubation. DAPI (blue), Endo/lysosome (green), Luc^{Cy5}mRNA@LNPs (red), scale bar = 20 μ m.

Reviewer #2 (Remarks to the Author): with expertise in cancer nanotherapy In this manuscript, the authors reported a unique strategy to reprogram tumor immune microenvironment by using mRNA nanoparticles to restore gasdermin B (GSDMB) N-terminal expression in tumor cells. The GSDMB N-terminal domain can induce pore formation on cell membrane, triggering the immunogenic cell death by releasing of DAMPs and cytokines. These mRNA nanoparticles were well characterized in vitro and in vivo. The therapeutic efficacy of GSDMB(NT) mRNA NPs was systematically examined in multiple tumor models in combination with anti-PD1 immune checkpoint blocker, as well as the change of tumor immune microenvironment. Overall, this work provides a new promising strategy for turning 'cold' tumors 'hot'. Below are several concerns that the authors need to further address. We appreciate the reviewer's valuable evaluation. We have addressed the comments below.

1. Several gasdermin family proteins can induce pore formation. The choice of gasdermin B in this work needs to be discussed.

Response: We are grateful to the reviewer for this valuable feedback. It has been shown that multiple gasdermin-N domains can induce pyroptosis owing to their pore-forming activity (Nature, 2016, doi:10.1038/nature18590). In comparison to other gasdermin-family members, the N-terminal domain of GSDMB exhibits a more significant pyroptotic response. Taking inspiration from this, the present study primarily focuses on development and efficacy investigations of mRNA-based nanotherapeutics that elicit the production of the GSDMB N-terminal domain, leading to pyroptosis-induced antitumor immunity that can synergize with checkpoint blockade for enhanced cancer therapy. We have included the explanation in the Discussion section on page 15, lines 337-339 in the revised manuscript.

2. The GSDMB(NT) mRNA NPs could also be taken up by immune cells (e.g., macrophages and DCs) in the tumors after i.t. injection. Will this induce immune cell toxicity and how could this impact the immune microenvironment change?

Response: We appreciate the reviewer for this comment.

Based on the data depicted in **Fig. 7c**, a pronounced increase in the populations of immune cells, including CD4⁺ T cells, CD8⁺ T cells, DCs, and NK cells, was evident in tumor tissues obtained from mice treated with GSDMB^{NT} mRNA@LNPs, when compared to the blank LNP control. These findings suggest that GSDMB^{NT} mRNA@LNPs exhibit minimal toxicity toward immune cells.

The updated **Fig. 1a** demonstrates the antitumor immunity mechanism of GSDMB^{NT} mRNA@LNPs. Specifically, GSDMB^{NT} mRNA@LNP treatment induces pyroptosis directly, without the involvement of protease cleavage, in a localized manner. Pyroptotic cell death further initiates the expression of proinflammatory cytokines, induces ICD, activates and recruits immune cells in the tumor microenvironment, thereby establishing a positive feedback loop for antitumor immunity. We have provided an explanation regarding the impact of GSDMB^{NT} mRNA@LNPs on the immune microenvironment in the Introduction section (page 4, lines 80-89) in the revised manuscript.

3. The anti-tumor effect of GSDMB(NT) mRNA NPs is attributable to the anti-tumor immune response, or the intrinsic pyroptosis/tumor suppressor effect of GSDMB, or both?

Response: We thank the reviewer for this question. The GSDMB^{NT} mRNA@LNPs not only kill cancer cells directly by inducing GSDMB^{NT}-mediated pyroptosis, but the induced pyroptosis (a type of inflammatory cell death) can further elicit antitumor immunity through initiating the expression of proinflammatory cytokines, inducing ICD, as well as activating and recruiting immune cells within tumors. We have included the explanation in the Discussion section on page 15, lines 341-347 in the revised manuscript.

4. The release of proinflammatory cytokines into blood might raise some concern on systemic safety.

Response: We appreciate this reviewer for the comment.

(1) As shown in **Fig. 4a**, following a single administration of high-dose GSDMB^{NT} mRNA@LNPs, serum concentrations of TNF- α and IFN- γ peaked at 24 hours post-injection and subsequently exhibited a dramatic decrease at 72 hours post-injection. The results indicate that the elevated levels of cytokines in the bloodstream are transient.

(2) We have performed additional studies outlined below to evaluate the safety profile of GSDMB^{NT} mRNA@LNPs.

To assess the in vivo safety profile of GSDMB^{NT} mRNA@LNPs, we monitored body weight changes and performed histology of major organs as well as aminotransaminase analysis following the administration of LNPs. As illustrated in **Extended Data Fig. 14**, no significant differences in body weight were observed across the treatment groups. In addition, no obvious signs of organ damage were detected in the H&E staining (**Extended Data Fig. 19**). Quantitative determination of major liver function markers (ALT and AST) demonstrated that administered GSDMB^{NT} mRNA@LNPs did not induce any obvious hepatic dysfunction (**Fig. 5d**). We have included the relevant Results and Methods in the revised manuscript (page 11, lines 253-255; page 27, lines 612-616). **Fig R5 (a), (b), and (c)** have been included in **Extended Data Fig. 14, Extended Data Fig. 19, and Fig. 5d**, separately. All changes have been highlighted in yellow.

(3) Our previous study (ACS Nano, 2022, doi.org/10.1021/acsnano.2c07822) regarding LNP-based mRNA COVID-19 vaccines has validated the excellent biocompatibility of the developed LNPs, as evidenced by the extremely low expression level of IL-6 at the injection sites.

Taken together, our findings indicate a favorable safety profile of the developed LNPs.

Fig. R5 In vivo safety profile of GSDMB^{NT} mRNA@LNPs. (a) The body weight of mice was monitored every two days until the end of the experiments. (b and c) Major organs and blood samples were collected on the second day after the final injection for histological (b) and aminotransaminase analyses (c).

5. In the study with distant tumors, the authors need to explain what induced CRT expression in the untreated tumors and may further measure the GSDMB(NT) expression in them.

Response: We appreciate this reviewer for the comments.

(1) As shown in **Fig. 7b**, elevated levels of cytokines were observed in both tumor tissues and serum, indicating that GSDMB^{NT} mRNA@LNP-induced pyroptosis elicited both local immune responses within the treated tumors and systemic immune responses. Furthermore, GSDMB^{NT}

mRNA@LNP treatment triggers DC maturation in lymph nodes (**Fig. 7c**), priming a systemic antitumor T-cell response. This finding is consistent with the previous study performed by BioNTech and Sanofi (Science Translational Medicine, 2021, doi/10.1126/scitranslmed.abc7804), which reported that mRNA-encoded cytokines predominantly act in a locoregional fashion, whereas the T cells and NK cells activated by the cytokines in situ confer systemic protective immunity. Therefore, we concluded that GSDMB^{NT} mRNA@LNPs evoke a systemic antitumor response in addition to the local immune responses within the treated tumors, leading to the upregulation of CRT expression in untreated lesions.

(2) We appreciate this reviewer for the suggestion regarding the measurement of the GSDMB(NT) expression in untreated tumors. GSDMB^{NT} mRNA@LNPs were exclusively administered to the tumor on the left flank, leading to local production of GSDMB^{NT} within the injected tumor microenvironment. Despite the absence of GSDMB^{NT} expression in the untreated lesions due to the localized administration route, the intratumoral upregulation of GSDMB^{NT} elicited pyroptosis, followed by a cascade of immunogenic events that promoted a robust systemic antitumor immune response.

6. The authors claimed that "However, even mRNAs encoding multiple cytokines still failed to induce successful antitumor immunity". The authors should be careful with the claim and at least cite references to support that.

Response: We thank this reviewer for the kind suggestion. We have made changes to this sentence in the Discussion section (page 18, lines 400-404). The changes have been highlighted in yellow.

Reviewer #3 (Remarks to the Author): with expertise in cancer immunology, immunogenic cell death

In the manuscript, Li et al. developed an elegant approach to trigger pyroptosis by mRNA lipid nanoparticle in order to sensitize immunologically “cold” tumors to checkpoint immunotherapy. The authors developed mRNA lipid nanoparticles (LNPs) encoding only the N-terminus of gasdermin to trigger pyroptosis. In the next part of the work, several mouse models have been used to analyze anti-tumor immunity. Additionally, the authors demonstrate that mRNA-mediated pyroptosis sensitized tumors to anti-PD-1 immunotherapy.

This is an interesting and clinically relevant approach. However, I strongly believe that in its current version, the manuscript is not mature enough to be published in Nature Communications. However, I suggest several key experiments which will help the authors to generate a revised version with justified and straightforward conclusions.

We express our gratitude to this reviewer for finding our work interesting and providing professional suggestions. To address the raised concerns, we have added additional data, which is described below.

Major comments

1. Annexin V/propidium iodide (PI) staining is not specific for apoptosis detection (Fig. 2). It is a general cell death detection technique that can identify the stage of cell death: AnV+PI-: early-stage and AnV+PI+ late stages. It is very well known that PS exposure can occur also in other cell death modalities such as necroptosis. Therefore, it is not a specific cell death marker/technique. The authors should also correct the terminology accordingly and it is necessary to show data on the early and late stages of cell death.

Response: We would like to thank this reviewer for these thoughtful and detailed comments.

(1) To evaluate the pyroptosis-induced cell death, we conducted an LDH release assay to investigate the lethal effect of GSDMB^{NT} mRNA@LNPs in different types of cells. As shown in **Fig. R1**, GSDMB^{NT} mRNA@LNP treatment led to markedly increased cell death rates of 51.5%, 47.5%, 32.6%, and 23.3% for HEK 293, HeLa, 4T1, and B16F10 cells, respectively, compared to the control groups (naked GSDMB^{NT} mRNA or blank LNPs) which exhibited substantially lower rates of below 5% for all four cell types. We have included the relevant Methods and Results in the revised manuscript (page 7, lines 146-151; page 22, lines 501-502). **Fig. R1** has also been included in **Fig. 2** as **Fig. 2b**. All changes have been highlighted in yellow.

(2) We thank this reviewer for the terminology suggestion. As suggested, we have made changes to the terminology accordingly in the revised manuscript. The data regarding the early and late stages of cell death has been included in **Fig. R6** (the updated **Fig. 2c** in the revised manuscript).

Fig. R1 LDH release-based cell death assay in HEK 293, HeLa, 4T1, and B16F10 cells after treatment with naked GSDMB^{NT} mRNA, LNPs, or GSDMB^{NT} mRNA@LNPs, respectively. Data are presented as means \pm SD (n = 3). Statistical significance was calculated via one-way ANOVA.

Fig. R6 Flow cytometry analysis of cells positive for propidium iodide and annexin V. All data are presented as means \pm SD ($n = 3$). Untreated cells served as the control (Ctrl) in all experiments.

2. The authors should provide clear experimental evidence that GSDMBNT mRNA@LNP induces pyroptosis (and not for example apoptosis and especially these should be shown for 4T1 and B16F10 cells) because in the current version of the manuscript, this experimental evidence is lacking. Moreover, it is necessary to discriminate this cell death induced by GSDMBNT mRNA@LNP from apoptosis and necroptosis.

Response: We appreciate this reviewer for the comments.

It has previously been reported that GSDM proteins are cleaved by caspases to release the GSDM-N domain, which permeabilizes the cell membrane, converting noninflammatory apoptosis to pyroptosis (Nature, 2020, doi.org/10.1038/s41586-020-2071-9). Pyroptosis describes a novel form of proinflammatory regulated cell death, characterized by cell swelling, plasma membrane bubbling, secretion of inflammatory cytokines (IL-1 β and IL-18) and lysis with release of inflammatory intracellular contents (Journal of cellular physiology, 2021, **236**, 7159-7175; PNAS, 2008, doi.org/10.1073/pnas.0707370105). As shown in **Fig. 2a** and **Extended Data 6-7**, pyroptotic morphological changes involving cytoplasmic swelling and cell membrane ballooning were observed in cells treated with GSDMB^{NT} mRNA@LNPs. Moreover, **Fig. 2b, 3b**

and **3c** show a substantial upregulation in released cell contents (including LDH, HMGB1 and ATP) in the GSDMB^{NT} mRNA@LNP treatment group in comparison to the control groups (naked GSDMB^{NT} mRNA or blank LNPs). These findings are consistent with existing studies on pyroptosis (Nature, 2020, doi.org/10.1038/s41586-020-2079-1; Nature Communication, 2023, doi.org/10.1038/s41467-023-36550-9). Taken collectively, we concluded that GSDMB^{NT} mRNA@LNPs can trigger pyroptosis. We have also highlighted the “in vitro evaluation of pyroptosis” portion (page 22, lines 490-510) in the Methods section in the revised manuscript.

3. What about the IL-1beta release? Can it be detected in the SN of dead/dying cells? This should be experimentally demonstrated.

Response: We thank this reviewer for the comments. As suggested, we performed an ELISA assay to measure IL-1 β in the supernatant of tumor cells transfected with GSDMB^{NT} mRNA@LNPs; unfortunately, no significant increase in IL-1 β release was observed when compared to control groups (naked GSDMB^{NT} mRNA or LNPs-treated). However, elevated IL-1 β levels were observed in both tumor tissues and serum in 4T1 tumor-bearing mice treated with GSDMB^{NT} mRNA@LNPs (**Fig. 5c**). One possible reason may be due to the fact that cytokine production in immune cells involves complex mechanisms (Cells 10, 111 (2021); Nature Reviews Immunology, volume 19, pages 205–217 (2019)); in contrast, tumor cells may lack some of these regulatory mechanisms, leading to insufficient cytokine production. As such, tumor cells may release cytokines in vitro at a level lower than the detectable assay range.

4. Surface-exposed calreticulin (CRT) was measured by confocal microscopy (Fig. 3). The description of how it was done is lacking in the Methods section. Of note that CRT exposure can be done only on flow cytometry and it should be analyzed on PI- cell population. Now, the authors demonstrate unspecific intracellular staining, and this could not be interpreted as surface exposure.

Response: Thank you for the comments.

(1) The experimental description regarding immunofluorescence imaging analysis of CRT expression has been included in the Methods section (page 23, lines 520-533) of the manuscript. As for the description of confocal microscopy imaging results, we have changed "CRT surface exposure" to "CRT expression measured by confocal microscopy" in the revised manuscript.

(2) As suggested, we measured CRT surface exposure on PI- cell population using flow cytometry. As shown in **Fig. R7** below, cells treated with GSDMB^{NT} mRNA@LNPs exhibited more CRT exposure than that of control groups (naked GSDMB^{NT} mRNA or blank LNPs). We have included the relevant Methods and Results in the revised manuscript (Page 8, lines 170-172; Page 24, lines 533-536). **Fig. R7** has also been included in **Extended Data Fig. 10**. All changes have been highlighted in yellow.

Fig. R7 GSDMB^{NT} mRNA@LNP treatment increases calreticulin (CRT) surface exposure of HEK 293 (a and e), HeLa (b and f), 4T1 (c and g), and B16F10 (d and h) cells. Cells were treated with the indicated treatments for 48 hours, and then collected and stained with an Alexa Fluor 488-labeled CRT antibody for flow cytometry analysis. All data are presented as means \pm SD (n = 3). Statistical significance was calculated via one-way ANOVA. Untreated cells served as the control (Ctrl) in all experiments. MFI: mean fluorescence intensity.

5. Analysis of the release of ATP and HMGB1 should be done in parallel with cell death analysis (See Fig. 3). Sometimes ATP can be released premortem before the appearance of the ruptured plasma membrane.

Response: We appreciate this reviewer for the comment. In addition to the analysis of ATP and HMGB1 release, we have performed a calcein-AM release assay that is commonly used to determine cell death resulting from pyroptosis-induced cell lysis (Science, 2020, doi.org/10.1126/science.aaz7548). For your convenience, we have highlighted the relevant experimental description in the Methods section (page 22, lines 502-510) in the revised manuscript.

6. Since the authors claim that this is pyroptosis IL-1 β and IL-18 should be also analyzed in the tumor and serum of the mice experiments (see Fig. 4).

Response: We appreciate this reviewer for the valuable suggestion. As suggested, the expression levels of IL-1 β and IL-18 were assessed in both tumor and serum samples from 4T1 tumor-bearing mice receiving the treatments indicated. **Fig. R8** shows that both GSDMB^{NT} mRNA@LNP treatment and combination therapy led to a significant increase in the intratumoral and serum expression levels of IL-1 β and IL-18, with higher expression observed in the group receiving the combination therapy treatment. However, no significant differences were found in the expression levels of the two cytokines between the control groups (PBS or blank LNPs) and

the a-PD1 monotherapy group. We have made changes in the Results section accordingly (Page 11, lines 246-248), with changes highlighted in yellow. The results are presented below in **Fig. R8** (please see the updated **Fig. 5c** in the revised manuscript).

Fig. R8 IL-1 β and IL-18 concentrations were measured in 4T1 tumor tissues or serum by ELISA.

Results are presented as means \pm SD (n = 4 mice per group).

7. Again, the authors used confocal microscopy to demonstrate CRT exposure at the cell surface (Fig 5). It is not specific because it does not discriminate from the intracellular staining. Another approach should be used to confirm this conclusion. The same is true for HMGB1 which the authors analyzed by WB. It shows just an increase in expression and has nothing to do with the extracellular release. Please use other techniques to demonstrate HMGB1 release in vivo. For example, one can measure it in the blood/serum.

Response: We thank this reviewer for these insightful suggestions.

(1) As suggested, CRT surface exposure was analyzed in 4T1 tumors and B16F10 tumors using flow cytometry. **Fig. R9** illustrates that the exposure of CRT was increased in the combination treatment group in comparison to the PBS-treated group. We have included the relevant Results in the revised manuscript (Page 11, lines 239-243; Page 13, lines 281-283;

Page 24, lines 533-536). **Fig. R9** has also been included in **Extended Data Fig. 12** and **15**. All changes have been highlighted in yellow.

Fig. R9 In vivo evaluation of CRT surface exposure in cells isolated from tumor tissues obtained from 4T1- (**a** and **b**) and B16F10-bearing (**c** and **d**) mouse models, respectively. Data are presented as mean \pm SD ($n = 3$ mice per group). MFI: mean fluorescence intensity.

(2) As suggested, the release levels of HMGB1 were assessed by ELISA in both tumor and serum samples from 4T1 tumor-bearing mice receiving the treatments indicated. Both GSDMB^{NT} mRNA@LNP treatment and combined treatment of GSDMB^{NT} mRNA@LNPs and a-PD1 exhibited elevated levels of HMGB1 in tumor and serum samples (**Fig. R10**). **Fig. R10** has also been included in the updated **Fig. 5c**. We have highlighted the changes (page 11, lines 246-248) in yellow in the revised manuscript.

Fig. R10 ELISA analysis of HMGB1 in tumor (left) and serum (right) samples from 4T1 tumor-bearing mice receiving the treatments indicated. Data are presented as means \pm SD (n = 4 mice per group).

8. From the presented data it is not clear whether GSDMB^{NT} mRNA@LNP-induced pyroptosis can be actually induced in vivo. It is necessary to demonstrate the cell death rate and type in vivo for 4T1 and B16F10 cellular models.

Response: We thank this reviewer for these comments.

(1) It is believed that pyroptosis is an inflammatory form of immunogenic cell death, accompanied by the extracellular release of pro-inflammatory cytokines, notably interleukin-1 β (IL-1 β) and interleukin-18 (IL-18) (Journal of cellular physiology, 2021, **236**, 7159-7175; PNAS, 2008, doi.org/10.1073/pnas.0707370105; Nature, 2020, doi.org/10.1038/s41586-020-2071-9). To investigate whether GSDMB^{NT} mRNA@LNPs induce pyroptosis in vivo, we evaluated the levels of pyroptosis-related inflammatory cytokines (IL-1 β and IL-18) in both tumor and serum samples from 4T1 tumor-bearing mice receiving the treatments indicated. The results shown in **Fig. R8** demonstrate that when compared to control groups (PBS or blank LNPs), GSDMB^{NT} mRNA@LNP treatment leads to a remarkable increase in the intratumoral and serum expression levels of IL-1 β and IL-18. We further investigated the expression of ICD biomarkers

(CRT and HMGB1) induced by GSDMB^{NT} mRNA@LNP-triggered pyroptosis. The data reported herein demonstrate a pronounced augmentation in the population of CRT+ cells (**Fig. 5a** and **6f**, **Extended Data Fig. 12** and **15**) and a concomitant elevation in the release of HMGB1 (**Fig. 5c** and **6e**) in tumor tissues obtained from 4T1 or B16F10 tumor-bearing mice that received treatment with GSDMB^{NT} mRNA@LNPs. We have highlighted the “in vivo evaluation of pyroptosis” portion (page 26, lines 594-608) in the Methods section in the revised manuscript.

(2) As suggested, to investigate the cell death rate in vivo, we harvested tumor tissues from B16F10 tumor-bearing mice that received either PBS or a combination treatment of GSDMB^{NT} mRNA@LNPs and a-PD1, in order to evaluate the population of PI-positive cells using flow cytometry analysis. As shown in **Extended Data Fig. 16** below, the proposed combination therapy results in a higher extent of cell death in tumor tissues than the PBS control. It is worth noting that cell pyroptosis of less than 15% induced by GSDMB^{NT} mRNA@LNPs in combination with a-PD1 was sufficient to elicit antitumor immunity (**Fig. 4c-e** and **Fig. 6b-d**). Our findings are consistent with the results reported in previous studies (Nature, 2020, doi.org/10.1038/s41586-020-2079-1). We have included the relevant Results and Methods in the revised manuscript (page 13, lines 283-285; page 27, lines 609-611).

Fig. R11 Cell viability assay in B16F10 tumors following combination treatment of aPD1 and GSDMB^{NT} mRNA@LNPs. (a and b) Assessment of the population of PI-positive cells. (c) Quantitative analysis of live cells in B16F10 tumors. Data are presented as means \pm SD (n = 3 mice per group).

9. The authors should show that antigens derived from pyroptotic cells can be presented by APC and can induce specific CD8 T cell responses. One can use OVA and OT-1 and OT-2 transgenic mice models.

Response: We appreciate the reviewer for this suggestion. It has been reported that intratumoral expression of GSDME^{NT} resulted in a significant increase in the number of tumor-specific CD8⁺ T cells within tumor tissues isolated from 4T1 tumor-bearing mice, indicating that GSDME^{NT}-triggered pyroptosis can induce specific CD8⁺ T cell responses (Nature, 2020, doi.org/10.1038/s41586-020-2071-9). Our results show that GSDMB^{NT} mRNA@LNP-triggered pyroptosis induces BMDC maturation and activation (please see more details in comment #10). In B16F10 tumor-bearing mice, GSDMB^{NT} mRNA@LNP treatment led to elevated expression of DC maturation biomarkers (CD11c and MHC-II) in lymph nodes and increased tumor-infiltrating CD8⁺ T cells within tumor tissues (**Fig. 7c**), indicating DC maturation and CD8⁺ T cell infiltration. The observed maturation of DCs, tumor infiltration of CD8⁺ T cells and antitumor immunity are thought to be a result of the presentation of antigens derived from pyroptotic cells by APCs and the subsequent induction of specific CD8⁺ T cell responses.

This present study aims to establish proof-of-concept for enhancing immunotherapy in preclinical “cold” tumor models using pyroptosis-triggering mRNA nanomedicines. Our current results show that GSDMB^{NT} mRNA@LNP-mediated pyroptosis can elicit antitumor immunity and reinforce aPD-1-mediated immunotherapy by reprogramming the TME from an immunosuppressive to an immunostimulatory state. Our future research will be focused on

exploring the mechanisms of the pyroptosis-induced cancer immunity cascade cycle, including antigen presentation, specific CD8⁺ T cell responses, and antitumor immunity.

10. In addition, it is also necessary to demonstrate the uptake of pyroptotic cells in vitro by bone-marrow-derived dendritic cells and to analyze their maturation/activation (MHC-II, CD80, CD86, production of pro-inflammatory cytokines) of dendritic cells.

Response: We appreciate this reviewer for the comment. As suggested, to explore the immune stimulation of DCs induced by GSDMB^{NT} mRNA@LNP-mediated pyroptosis, B16F10 cells were pretreated with PBS (Ctrl), naked GSDMB^{NT} mRNA, blank LNPs or GSDMB^{NT} mRNA@LNPs, followed by coculture with BMDCs obtained from female C57BL/6 mice as previously reported [Nature Communications, 2023, doi.org/10.1038/s41467-023-36550-9]. The levels of pro-inflammatory cytokines (IFN- γ , IL-1 β and TNF- α) in the culture medium were analyzed by ELISA, and flow cytometry was used to measure the expression of maturation biomarkers (MHC-II and CD86) on DCs. As shown in **Fig. R12** below, pretreatment with GSDMB^{NT} mRNA@LNPs increased production of IFN- γ , IL-1 β and TNF- α by 3.2-, 8.1-, and 9.6-fold, respectively, over PBS pretreatment (Ctrl). Additionally, coculture with GSDMB^{NT} mRNA@LNP-pretreated B16F10 cells resulted in a significant upregulation of CD86 and MHC-II surface expression on BMDCs by 4.1- and 1.8-fold, respectively, compared to the control group. Taken together, pyroptotic tumor cells efficiently induce the maturation of DCs via GSDMB^{NT} mRNA/LNP-mediated pyroptosis. We have included the relevant Methods and Results in the revised manuscript (Page 8, lines 175-183; Page 23, lines 511-519). **Fig. R12** has also been included in **Fig. 3d** and **e**. All changes have been highlighted in yellow.

Fig. R12 Immune stimulation of BMDCs by GSDMB^{NT} mRNA/LNPs. B16F10 cells were pretreated with PBS (Ctrl), naked GSDMB^{NT} mRNA, blank LNPs or GSDMB^{NT} mRNA@LNPs, followed by coculture with BMDCs for 48 hours. (a) Quantitative determination of proinflammatory cytokines IFN-γ, IL-1β and TNF-α using ELISA assay. Data are presented as means ± SD (n = 3). Statistical significance was calculated via one-way ANOVA. (b) Analysis of DC maturation biomarkers (MHC-II and CD86) using flow cytometry.

11. One of the golden standards to demonstrate that indeed the given cell death type is ICD one should use the tumor prophylactic vaccination model. It is strongly advised to add such data to the revised version of the manuscript.

Response: We appreciate the kind comment you provided and we agree with your suggestion. Recent studies have validated that N-terminal domain gasdermin-induced pyroptosis represents a form of ICD by use of a B16 tumor prophylactic vaccination model against tumor cell

rechallenge (Nature, 2020, doi.org/10.1038/s41586-020-2071-9; Nature Communications, 2023, doi.org/10.1038/s41467-023-36550-9). Other studies have also demonstrated that ICD involves the release of DAMPs (e.g. CRT and HMGB1) and the resulting establishment of immunological memory (Science Translational Medicine, 2021, doi: 10.1126/scitranslmed.aba9772; Nature Cancer, 2020, doi.org/10.1038/s43018-020-0095-6). Based on these findings, we conducted a tumor rechallenge experiment described in previous studies (Science Translational Medicine, 2021, doi: 10.1126/scitranslmed.aba9772; Nature Cancer, 2020, doi.org/10.1038/s43018-020-0095-6) in combination with our observations of released DAMPs and inflammatory cytokines to investigate the involvement of ICD in the LNP-mediated anticancer immune response. The results showed that mice treated with a combined treatment regimen of GSDMB^{NT} mRNA@LNPs and a-PD1 exhibited an elevated CRT surface exposure and HMGB1 release (**Extended Data Fig. 12 and 15, Fig. 5c and 6e**) and showed resistance to tumor rechallenge in the B16F10-bearing mouse model (**Extended Data Fig. 20**), which is indicative of the establishment of immunological memory. Our results are consistent with what has been reported in previous work and the above findings serve as supportive evidence of the induction of ICD through LNP-mediated pyroptosis.

We would like to reiterate that we agree with this reviewer's comments regarding the confirmation of the type of cell death induced by LNPs. Additionally, we are currently preparing for a tumor prophylactic vaccination model experiment aimed at further confirming ICD as described previously (Nature, 2020, doi.org/10.1038/s41586-020-2071-9; Journal for ImmunoTherapy of Cancer. 2020; 8(1): e000337). However, due to time and material constraints, we only included the data from the in vivo rechallenge experiment to demonstrate the establishment of immunological memory in the response letter. We sincerely hope that this reviewer will review our revised manuscript with this information. We wish to kindly state that this present work is primarily focused on developing an mRNA-based nanotherapeutic approach

as a proof-of-concept for enhanced ICB-mediated cancer immunotherapy by delivering mRNA encoding the GSDMB N-terminus to induce pyroptosis. In our future work, we intend to conduct a comprehensive study that includes a full elucidation of the mechanisms underlying pyroptosis-inducing LNPs for sensitizing tumors with ICB therapy.

We have included the Methods and Results regarding CRT surface exposure, HMGB1 release and tumor rechallenge experiments in the revised manuscript (page 8, lines 170-172; page 24, lines 533-536; page 11, lines 239-243; page 13, lines 281-283; page 24, lines 533-536; page 11, lines 246-248; page 13, lines 303-305; page 24, lines 547-550 have been updated. **Extended Data Fig. 10, 12, 15** and **Fig. 5c** have been included as mentioned above). **Fig. R13** has also been included in **Extended Data Fig. 20** (shown below). All changes have been highlighted in yellow.

Fig. R13 Enhanced immunological memory of GSDMB^{NT} mRNA@LNPs in combination with aPD-1 using a B16F10 tumor rechallenge model. (a) Experimental timeline for treatment of B16F10 tumor-bearing mice and s.c. rechallenge. B16F10 tumor-bearing mice that had previously received a combination treatment regimen of aPD-1 and GSDMB^{NT} mRNA@LNPs were rechallenged with 5×10^5 B16F10 cells on the left flank. Naive mice were subcutaneously implanted with the same number of B16F10 cells on day 0 to serve as a control. The volume of the rechallenged tumors was monitored every two days. (b) The tumor growth profile for the

naive group and the combination treatment group (GSDMB^{NT} mRNA@LNPs in combination with aPD-1). Data are presented as means \pm SD (n = 3 mice per group).

12. The authors should demonstrate that indeed pyroptosis is important for the sensitization of immunologically “cold” tumors to checkpoint immunotherapy. Adding some important controls (e.g., non-ICD) will re-enforce their conclusions.

Response: We appreciate the reviewer’s comments. In this study, we have developed an mRNA-based nanotherapeutic approach that delivers GSDMB^{NT} mRNA@LNPs to induce pyroptosis in tumor lesions, generating a favorable immunogenic “hot” tumor microenvironment and promoting antitumor immunity. The results presented in this study provide new insights into sensitizing immunologically “cold” tumors to checkpoint immunotherapy. In our future work, we are interested in exploring various mechanisms for enhancing the effectiveness of ICB-mediated immunotherapy, such as the use of non-ICD agents that synergistically improve T-cell responsiveness. We thank this reviewer for the suggestion to add some control groups (such as non-ICD). However, due to time limitations and logistical constraints regarding repeating experiments, we will include these explorations in our future study.

Minor comments

A description of the statistical methods should be indicated in each figure legend. The number of experiments should be also indicated in each figure legend. Now it is missing, e.g., see figure legend of Fig. 3.

Response: We appreciate this reviewer for the suggestion. We have included the description of the statistical methods and the number of experiments.

Reviewer #4 (Remarks to the Author): with expertise in pyroptosis, immunology

Inducing pyroptosis is considered to be a promising strategy of cancer treatment. In this manuscript, the author developed a general mRNA nanomedicine approach to turn the immunosuppressive cold tumors into inflammatory hot tumors. This conversion promotes the infiltration of immune cells in TME, enhancing the antitumor activity of anti-PD-1 immunotherapy. They proved that the pyroptosis triggering-mRNA is effective in many kinds of tumor cell as well as in vivo. In summary, the author discovered a novel pyroptosis triggering therapy, which can also enhance the efficacy of other immunotherapies. However, there are still some key issues needed to be addressed.

We appreciate the thorough evaluation and insightful suggestions provided by the reviewer. In order to address the important points raised, we have provided point-by-point responses below.

Major points

1. It would be better if the author could explain the reason for choosing GSDMB instead of other GSDM family members in this therapy? Is GSDMB-NT the most efficient one to induce pyroptosis? Besides, considering that GSDMB is not expressed in rodents, mouse cells may lack the mechanism of post-translational modification of GSDMB and inhibition of pore-forming activity compared with human cells. Therefore, it is questionable whether it is reasonable to study the function of GSDMB using mice as animal models. In addition, female mice were chosen to perform in vivo experiments in this study. Is there any difference in anti-tumor effect of LNP-GSDMB-NT between male and female mice?

Response: We appreciate this reviewer for the comments.

(1) We are grateful to the reviewer for the suggestion to explain the reason for choosing GSDMB in this study. It has been shown that multiple gasdermin-N domains can induce pyroptosis owing to their pore-forming activity (Nature, 2016, doi:10.1038/nature18590). In

comparison to other gasdermin-family members, the N-terminal domain of GSDMB exhibits a more significant pyroptotic response. Taking inspiration from this, the present study primarily focuses on the development and efficacy investigations of mRNA-based nanotherapeutics that elicit the production of the GSDMB N-terminal domain, leading to pyroptosis-induced antitumor immunity that can synergize with checkpoint blockade for enhanced cancer therapy. We have included the explanation in the Discussion section on page 15, lines 337-339 in the revised manuscript.

(2) Recent studies have shown that the cleaved N-terminal domain of GSDMB induces pyroptosis-mediated antitumor immunity in several tumor-bearing mouse models (Science, 2020, doi.org/10.1126/science.aaz7548; Nature, 2020, doi.org/10.1038/s41586-020-2071-9). Despite the absence of GSDMB orthologs in mice, human GSDMB (hGSDMB) can be cleaved into its N-terminal and C-terminal domains by mouse caspases/proteases (such as granzyme A). It has been clearly elucidated that the N-terminal domain of GSDMB binds membrane lipids (phosphoinositides and cardiolipin) and perforates the plasma membrane, thus exhibiting membrane-disrupting cytotoxicity and inducing pyroptosis in cells (Nature, 2016, doi:10.1038/nature18590). Drawing from the findings previously reported, we propose an mRNA-based nanotherapeutic approach as a proof-of-concept for enhanced cancer immunotherapy by directly delivering mRNA encoding the GSDMB N-terminus to induce pyroptosis, even without requiring cleavage. In this work, we evaluated GSDMB^{NT} mRNA@LNP-induced pyroptosis in murine cell lines (4T1 and B16F10) and observed comparable findings to those obtained in human cell lines (HEK 293 and HeLa). We expect that our strategy could provide new insights into developing pyroptosis-triggered mRNA nanomedicines for cancer treatment in future clinical practice.

(3) To facilitate cross-comparison with previously published findings in our experiments, we employed mouse models of the same sex that had been reported in studies related to

pyroptosis. (Nature, 2020, doi.org/10.1038/s41586-020-2079-1; Nature, 2020, doi.org/10.1038/s41586-020-2071-9). We obtained comparable results to those reported in the literature.

2. As a method of injection therapy of exogenous substances, the stability and persistence of exogenous GSDMB-NT mRNA in cells should be fully considered, because the activity of RNase in many tumor cells is relatively high. Meanwhile, the metabolism of the material, including ionizable cationic lipid (AA3-Dlin), phospholipid (DOPE), cholesterol, and PEG, used for mRNA transport, also need to be considered, so as to illustrate the efficiency and side effects of this strategy in a long-term use.

Response: We appreciate the reviewer for these comments.

(1) To investigate whether our LNPs protect mRNA from RNase degradation, mRNA@LNPs were incubated with RNase (50 ng/mL) for 0, 0.5, 1.0, 2.0, and 4.0 hours, and then were subjected to Gel-red-infused 1% agarose gel electrophoresis. As shown in **Fig. R14** below, there was no observed mRNA degradation after LNP encapsulation, while naked mRNA quickly degraded in RNase, which is consistent with previous studies (NPJ Vaccines, 136 (2022)).

Fig. R14 Stability of naked mRNA and LNP-encapsulated mRNA against RNase.

(2) The four components in LNP formulation have been determined to be biocompatible. DOPE, cholesterol, and PEG have been extensively used in COVID-19 mRNA vaccines produced by Moderna and Pfizer-BioNTech. In their assessment reports, the pharmacokinetic properties of DOPE, cholesterol and PEG have been extensively analyzed. Data show that cholesterol and DOPE are metabolized and excreted in the same way as the endogenous lipids, and lipid-PEG is metabolized by hydrolytic metabolism of the ester and amide functionalities. More details can be found in the literature below: Assessment Reports of mRNA-1273 (Moderna) and BNT162b2 (Pfizer-BioNTech) vaccines; *New England Journal of Medicine*, 383(20), 1920-1931; *New England Journal of Medicine*, 383(25), 2439-2450; *Nature* volume 586, pages589–593 (2020). The design, synthesis, and characterization of the novel ionizable lipid AA3-DLin and its LNP formulations have been fully studied (*ACS Nano*, 2022, doi.org/10.1021/acsnano.2c07822). AA3-DLin is designed to incorporate biodegradable linkers connecting the headgroups with hydrocarbon chains, which facilitates safe clearance after mRNA delivery. The distribution, metabolism and pharmacokinetics (DMPK) of AA3-DLin remain unexplored; but the DMPK of a close structural analogue DLin-MC3-DMA (which is a major lipid excipient of Onpattro®) has been extensively studied and the data show efficient hydrolysis and clearance (Assessment report of Onpattro®). In this work, our results show that mice treated with the developed AA3-DLin LNPs did not exhibit any significant weight loss, abnormalities in major organs, or obvious hepatic dysfunction (**Extended Data Fig. 14, Extended Data Fig. 19, and Fig. 5d**), demonstrating that the developed AA3-DLin LNPs possess good mRNA delivery efficacy and biocompatibility. In the future, we will further investigate the DMPK properties of AA3-DLin to facilitate the translation of our research regarding mRNA/LNP-induced pyroptosis immunotherapy into preclinical trials.

3. Initially, the authors examined the stability of GSDMB-NT mRNA@LNPs in PBS and cell culture medium. However, tumor microenvironment is a special physiological/pathological

environment, and its pH and esterase activity may affect the stability of lipid nanoparticles. Therefore, it is necessary to re-explore GSDMB-NT mRNA@LNPs stability in some typical tumor extreme microenvironments.

Response: We appreciate this reviewer for the comment. As suggested, the stability of the developed GSDMB^{NT} mRNA@LNPs was evaluated by monitoring particle sizes in various culture media, including pH 6.5 buffer (to mimic the weakly acidic pH of many tumors), pH 7.4 buffer (to mimic the normal physiological pH), 10% or 20% plasma (to mimic esterase-enriched conditions). GSDMB^{NT} mRNA@LNPs exhibit no obvious changes in particle sizes when incubated under the above conditions, implying good stability within acidic or esterase-enriched pathological tumor microenvironments (**Fig. R16**). We have included the relevant Methods and Results in the revised manuscript (Page 6, lines 120-126; Page 20, lines 457-460). **Fig. R16** has also been included in **Fig. 1** as **Fig. 1d**. All changes have been highlighted in yellow.

Fig. R16 DLS measurement of GSDMB^{NT} mRNA@LNPs under the conditions indicated, including pH 6.5 buffer, pH 7.4 buffer, 10% or 20% plasma.

4. As human tumors are much larger than those in mouse models, the treatment of GSDMB-NT mRNA@LNPs injected into human tumor lesions should also require more precisely. Is it better to be near the outside of the tumor, favoring the release of cytokines into the surrounding

environment to elicit stronger anti-tumor immunity, or is it better to be inside the solid tumor, acting more thoroughly and uniformly from the inside out across the tumor area. Most importantly, intra-tumoral administration is widely considered as impractical in clinical and four-dose usage adds troubles for its application.

Response: We appreciate this reviewer's comments.

(1) We refer to the established intratumoral injection protocol for guidance in performing the intratumoral injection. Briefly, to make sure the treatment solution is distributed to the tumor lesions uniformly, the needle was inserted into the tumor lesions from a single puncture entry point. After injecting $\frac{1}{4}$ volume of the solution, the needle was slowly withdrawn a short distance (while keeping the needle within the tumor lesions), then the remaining solution was injected into multiple points in the same tumor lesions using the above method.

(2) The intratumoral administration of mRNA-based nanotherapeutics has been widely utilized for the treatment of certain types of cancer. For example, BNT131 (SAR441000) is an intratumoral therapeutic developed by BioNTech/Sanofi for treating solid tumors. It involves direct injection of an mRNA cancer vaccine encoding the cytokines IL-12sc, IL-15sushi, IFN α and GM-CSF into tumor tissues. BNT131 is currently being investigated in a Phase I trial in combination with a PD-1 inhibitor in patients with advanced solid tumors, including melanoma (NCT03871348). As another example, Moderna's mRNA-2752 is an intratumorally administered mRNA vaccine that utilizes lipid nanoparticles to encapsulate mRNAs encoding human OX40L, IL-23, and IL-36 γ . It is currently being evaluated in a Phase I study in patients diagnosed with solid tumor malignancies, such as breast cancer melanoma (NCT03739931).

(3) Our study aims to demonstrate a proof-of-concept for the design of pyroptosis-triggering mRNA/LNPs for eliciting antitumor immunity and provide a strategy for enhancing the sensitization of immunosuppressive tumors to ICB-mediated immunotherapy. We have

evaluated the safety profile of GSDMB^{NT} mRNA@LNPs. Our results show that mice treated with GSDMB^{NT} mRNA@LNPs did not exhibit any significant weight loss, abnormalities in major organs, or obvious hepatic dysfunction. In order to reduce the necessity of multiple dosing regimens, our future work will focus on developing targeted LNPs that can deliver self-amplifying GSDMB^{NT} mRNA (capable of long-term protein expression) to tumor tissues via systemic administration routes. However, these studies are beyond the scope of this manuscript.

5. As shown in Fig 3, the LNPs used in the current study are not tumor-specific and also target normal cells, resulting in killing of both targeted tumor and normal cells. Thus the strategy described in this study is not as “safe” as the authors mentioned in the discussion. It is better to modify the LNPs with appropriate groups to improve the specificity of LNPs.

Response: We appreciate this reviewer’s comment. The aim of this study is to establish a proof-of-concept for designing an mRNA-based nanotherapeutic approach that triggers pyroptosis to induce antitumor immunity and enhance the sensitization of tumors to ICB therapy. To reduce the possible side effects of GSDMB^{NT} mRNA@LNPs on normal tissues/cells, LNPs were intratumorally injected into the tumor tissues directly. The in vivo safety evaluation results demonstrate that intratumoral injected GSDMB^{NT} mRNA@LNPs do not show obvious damage to major organs and liver functions (**Extended Data Fig. 14, Extended Data Fig. 19, and Fig. 5d**). We are devoted to developing modified LNPs to target cancer cells specifically in our future work; however, this is beyond the scope of our current study.

Minor points

1. In figure 2D (corresponding description in line 122-125), what the treatment of GSDMB-NT mRNA group stands for? Was the mRNA added to the cell culture medium directly or transfecting cells with other method? It would be clearer if the author could describe this control group either in figure legend or in the methods part.

Response: We would like to extend our gratitude to the reviewer for bringing this to our attention. The term "GSDMB^{NT} mRNA" denotes the treatment where naked GSDMB^{NT} mRNA was added directly to the culture medium and incubated with the cells without the use of any transfection agent. In order to avoid any confusion, we have updated the designation of "GSDMB^{NT} mRNA" to "naked GSDMB^{NT} mRNA" for better clarity.

2. The authors need to confirm the expression of GSDMB-NT in LNPs targeted cells by immunoblot analysis. Additionally, since the authors mentioned adverse clinical reactions were not observed in the mouse models, these data need to be included in the manuscript.

Response: We thank this reviewer for the suggestion.

(1) As suggested, we performed Western blots to determine the expression level of GSDMB^{NT} in HEK293 cells transfected with GSDMB^{NT} mRNA@LNPs. As shown in **Fig. R17** below, cells transfected with GSDMB^{NT} mRNA@LNPs showed GSDMB^{NT} protein expression, while no GSDMB^{NT} signal was observed in cells transfected with naked GSDMB^{NT} mRNA. We have included the relevant Methods and Results in the revised manuscript (Page 7, lines 140-141; Page 21, lines 471-481). **Fig. R17** has also been included in **Extended Data Fig. 5**. All changes have been highlighted in yellow.

Fig. R17 Western blot analysis of GSDMB^{NT} expression

(2) During the experimental period, tumor growth, body weight and deaths were monitored in 4T1- or B16F10-bearing mouse models. The results have been included in **Fig. 4c-e**, **Fig. 6b-d** and **Extended Data Fig. 14** in the revised manuscript.

REVIEWERS' COMMENTS

Reviewer #1 (Remarks to the Author):

The authors have addressed my comments.

Reviewer #2 (Remarks to the Author):

The revised manuscript has adequately addressed most of the reviewer's concerns. Two minor points the authors may further clarify: i) to clearly understand the effect of GSDMB(NT) mRNA NPs on macrophages/DCs, an in vitro cytotoxicity experiment is recommended; ii) how the systemic immune responses induced CRT expression in untreated tumors should be explained.

Reviewer #3 (Remarks to the Author):

The authors generated an improved version of the manuscript in which in my opinion conclusions are very straightforward and very innovative. Most of the comments are addressed by the authors. Overall the manuscript will be of great interest to the scientific community working on immunogenic cell death, pyroptosis, and cancer immunotherapy in general.

Comment:

One remaining comment is related to Figure R12B. It is strongly advised to provide statistical analysis of dot plots (bar chart, quantitative data of MHC-II+ and CD86+) and to indicate the number of repeats in the legend.

Reviewer #4 (Remarks to the Author):

The authors have addressed all my questions. I'd like to support its publication.

Reviewer #1 (Remarks to the Author):

The authors have addressed my comments.

Response: We thank this reviewer for the positive comments.

Reviewer #2 (Remarks to the Author):

The revised manuscript has adequately addressed most of the reviewer's concerns. Two minor points the authors may further clarify: i) to clearly understand the effect of GSDMB(NT) mRNA NPs on macrophages/DCs, an in vitro cytotoxicity experiment is recommended; ii) how the systemic immune responses induced CRT expression in untreated tumors should be explained.

Response: We thank this reviewer for the positive comments.

(1) As suggested, we performed a Cell Counting Kit-8 (CCK-8) assay to evaluate the cytotoxicity of GSDMB^{NT} mRNA@LNPs on bone marrow-derived macrophages/DCs obtained from C57BL/6 mice. As shown in **Fig. R18** below, GSDMB^{NT} mRNA@LNPs showed limited cytotoxicity at a high concentration (with 1.5 $\mu\text{g}/\text{mL}$ GSDMB^{NT} mRNA). We have included the relevant Methods and Results in the revised manuscript (Page 9, lines 187-189; Page 23, lines 525-531). Data shown in **Fig. R18** has been included in the updated **Supplementary Fig. 12**. All changes have been highlighted in yellow.

Fig. R18 Cell viability of bone marrow-derived macrophages and DCs after treatment with GSDMB^{NT} mRNA@LNPs for 48 hours. Data are presented as means \pm SD (n = 4 replicates).

Our study aims to present a proof-of-concept regarding the development of pyroptosis-triggering mRNA/LNPs for eliciting antitumor immunity and provide a strategy for enhancing the sensitization of immunosuppressive tumors to ICB-mediated immunotherapy. The safety profile of GSDMB^{NT} mRNA@LNPs was assessed and indicated that mice administered with GSDMB^{NT} mRNA@LNPs did not show significant weight loss, abnormalities in major organs, or evident hepatic dysfunction. To further enhance the safety of GSDMB^{NT} mRNA@LNPs, our future investigations will be focused on the development of targeted LNPs capable of delivering GSDMB^{NT} mRNA to tumor tissues through systemic routes of administration.

(2) Previous studies have demonstrated that the induction of ICD after local therapy can provoke a systemic antitumor T-cell response to kill distant tumor cells (Science Translational Medicine, 2021, doi/10.1126/scitranslmed.abc7804; Angew. Chem. Int. Ed. 2019, doi.org/10.1002/anie.201804882). As shown in **Fig. 8d**, CD8⁺ T cell infiltration and recruitment were observed in untreated tumor tissues after local treatment of GSDMB^{NT} mRNA@LNPs. The infiltrated CD8⁺ T cell initiated cellular immunity and resulted in tumor cell death which is accompanied by the release of tumor antigens and CRT.

Reviewer #3 (Remarks to the Author):

The authors generated an improved version of the manuscript in which in my opinion conclusions are very straightforward and very innovative. Most of the comments are addressed by the authors. Overall the manuscript will be of great interest to the scientific community working on immunogenic cell death, pyroptosis, and cancer immunotherapy in general.

Comment:

One remaining comment is related to Figure R12B. It is strongly advised to provide statistical analysis of dot plots (bar chart, quantitative data of MHC-II⁺ and CD86⁺) and to indicate the number of repeats in the legend.

Response: We thank this reviewer for the positive comment. As suggested, we have included the statistical analysis of dot plots for MHC-II⁺ and CD86⁺ in **Fig. R19** below (**Supplementary Fig. 11** in the updated Supplementary Information).

Fig. R19 Flow cytometry analysis of DC maturation biomarkers (MHC-II and CD86). Data are presented as means \pm SD (n = 3). Statistical significance was calculated via one-way ANOVA.

Reviewer #4 (Remarks to the Author):

The authors have addressed all my questions. I'd like to support its publication.

Response: We thank this reviewer for the positive comment.